# Unlocking Large-Scale Crop Field Delineation in Smallholder Farming Systems with Transfer Learning and Weak Supervision

Sherrie Wang [1,2,3,*], François Waldner [4] and David B. Lobell [3]

1    Goldman School of Public Policy, University of California, Berkeley, 2607 Hearst Ave, Berkeley, CA 94720, USA
2    Institute for Computational and Mathematical Engineering, Stanford University, 475 Via Ortega, Stanford, CA 94305, USA
3    Department of Earth System Science, Stanford University, 473 Via Ortega, Stanford, CA 94305, USA
4    European Commission Joint Research Centre, Via Enrico Fermi 2749, 21027 Ispra, Italy
*    Correspondence: sherwang@berkeley.edu

**Highlights:**

**What are the main findings?**
- We develop a method for accurate, scalable field delineation in smallholder systems.
- Fields are delineated with state-of-the-art deep learning and watershed segmentation.

**What is the implication of the main finding?**
- Transfer learning and weak supervision reduce training labels needed by 5× to 10×.
- 10,000 new crop field boundaries are generated in India and publicly released.

**Abstract:** Crop field boundaries aid in mapping crop types, predicting yields, and delivering field-scale analytics to farmers. Recent years have seen the successful application of deep learning to delineating field boundaries in industrial agricultural systems, but field boundary datasets remain missing in smallholder systems due to (1) small fields that require high resolution satellite imagery to delineate and (2) a lack of ground labels for model training and validation. In this work, we use newly-accessible high-resolution satellite imagery and combine transfer learning with weak supervision to address these challenges in India. Our best model uses 1.5 m resolution Airbus SPOT imagery as input, pre-trains a state-of-the-art neural network on France field boundaries, and fine-tunes on India labels to achieve a median Intersection over Union (mIoU) of 0.85 in India. When we decouple field delineation from cropland classification, a model trained in France and applied as-is to India Airbus SPOT imagery delineates fields with a mIoU of 0.74. If using 4.8 m resolution PlanetScope imagery instead, high average performance (mIoU > 0.8) is only achievable for fields larger than 1 hectare. Experiments also show that pre-training in France reduces the number of India field labels needed to achieve a given performance level by as much as 10× when datasets are small. These findings suggest our method is a scalable approach for delineating crop fields in regions of the world that currently lack field boundary datasets. We publicly release 10,000 Indian field boundary labels and our delineation model to facilitate the creation of field boundary maps and new methods by the community.

**Keywords:** agriculture; field delineation; segmentation; deep learning; transfer learning; weak supervision; remote sensing; smallholders

## 1. Introduction

A crop field is the basic unit of management in agriculture. Delineating field boundaries allows one to capture the size, shape, and spatial distribution of agricultural fields, which are important characteristics of rural landscapes [1,2]. By enabling field-level analysis, field boundaries are helpful inputs to crop type mapping [3–6], yield mapping [7–12], and digital

agriculture services [13]. Previous work has also related field size to productivity [14–16], pest and disease spread [17,18], and species diversity [19,20].

Despite their usefulness, field boundary datasets remain unavailable in most countries. Where they do exist, field boundaries are either gathered through ground surveys [21], submitted by farmers to a statistical agency [10,22], or extracted from very high resolution remote sensing imagery [23–25]. All of these methods are expensive, labor-intensive, and require sophisticated statistical infrastructure; as a result, datasets are most likely to exist in high-income countries. In many medium- and low-income countries, the capacity to conduct surveys or fly imaging aircraft is limited or nonexistent. To our knowledge, there is no large-scale dataset in any smallholder region in the Global South, despite the importance of agricultural management and productivity in these regions.

High-resolution satellite imagery and recent advances in computer vision offer opportunities for automated field boundary delineation at low cost. Researchers have demonstrated that delineation can be automated in industrial agricultural systems like those in North America [2,26,27], Europe [28–30], and Australia [31] using publicly-available Landsat or Sentinel-2 imagery. In some cases, previous methods used large quantities of historical field boundaries and supervised machine learning to prove automated delineation is possible. For example, Waldner and Diakogiannis [23] used a novel deep learning architecture to delineate fields accurately in South Africa using boundaries from a government agency. In other cases, historical data was unavailable even in high-income countries, and researchers collected their own data for the study. Waldner et al. [31] obtained over 70,000 field labels in Australia and used them to train a deep learning model that then delineated 1.7 million fields throughout the Australian grains zone. Still other methods circumvent the need for labeled boundaries by employing unsupervised methods. In the US, Yan and Roy [2] used edge detection and active contour segmentation to delineate fields in the US.

These advances in industrial agricultural systems offer reasons to be optimistic about smallholder field delineation. Indeed, a number of recent works have applied machine learning methods to smallholder field delineation with success. For example, Persello et al. [21] used very high resolution satellite imagery and labels collected via ground surveys to train a convolutional neural network (CNN) to predict field contours in Nigeria and Mali; Zhang et al. [32] used Sentinel-2 images, labels derived from unsupervised edge detection, and a recurrent residual U-Net to delineate fields in Heilongjiang province, China; and Estes et al. [33] used unsupervised edge detection algorithms, a random forest with active learning trained to classify cropland, and PlanetScope imagery to delineate fields throughout Ghana. However, gaps still exist between prior work and delivering large-scale, high-accuracy field boundaries. Persello et al. [21] were constrained by highly localized field boundary labels to study areas of a few square kilometers. Zhang et al. [32] worked at a larger spatial scale but used Sentinel-2 imagery (10 m resolution), which prevented small fields from being delineated. Estes et al. [33] also delineated boundaries at large scale, but supervised deep learning methods are known to produce higher accuracies than unsupervised edge detection algorithms or random forests. In order for automated field delineation to extend to smallholder systems, a number of challenges must be overcome.

The first major challenge to smallholder field delineation is the availability of satellite images at high enough resolution to see field boundaries. Figure 1 compares the appearance of fields in South Africa, France, and India in imagery taken by Landsat-8, Sentinel-2, PlanetScope, and Airbus SPOT satellites. Field boundaries in South Africa and France can be seen clearly in Landsat-8 (30 m) and Sentinel-2 (10 m) imagery, but smallholder fields in India require PlanetScope (4.8 m) or Airbus SPOT (1.5 m) imagery to be resolved (Figure 1). Historically, very high resolution satellite imagery was expensive to access and available only for a fraction of the Earth's surface. Only recently has the cost of access been reduced either via the launch of cubesats (PlanetScope) or user-friendly integration of high resolution basemaps (Airbus SPOT) in cloud-based platforms like Descartes Labs.

Geographic coverage has also expanded as more satellites have been launched and data storage has become cheaper. Two recent data developments are particularly notable: first, Planet partnered with Norway's International Climate & Forests Initiative to release PlanetScope mosaics across South Asia, Southeast Asia, and Sub-Saharan Africa [34]; and second, Descartes Labs added access to annual global Airbus OneAtlas Basemaps for users of the platform [35,36]. With the increasing accessibility of high resolution satellite imagery, now is the time to investigate their use for smallholder field delineation.

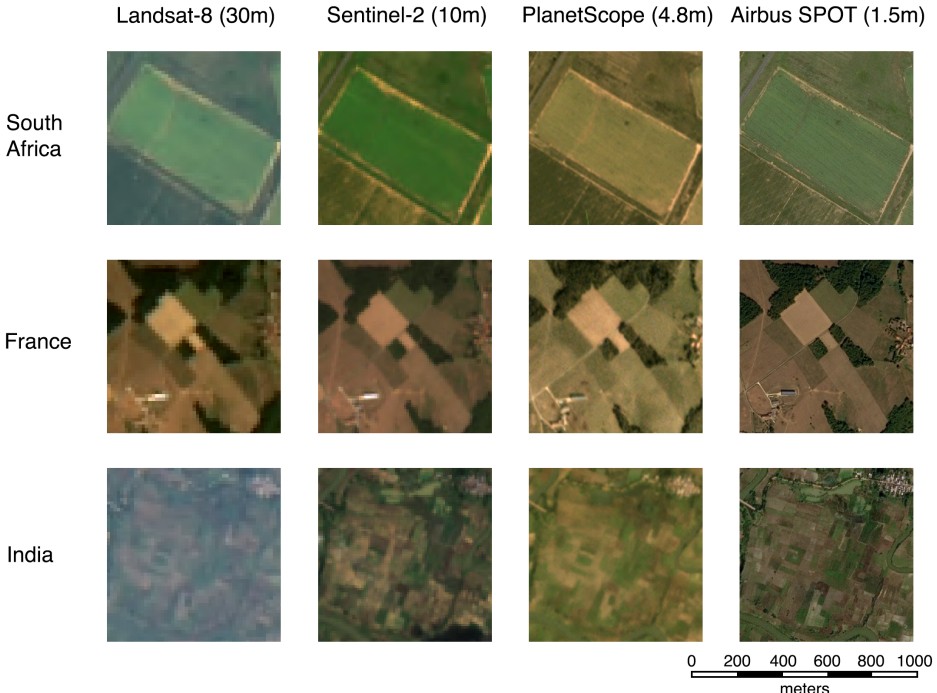

**Figure 1. Satellite images taken over agricultural areas in South Africa, France, and India.** From left to right, images are taken by Landsat-8 at 30 m resolution, Sentinel-2 at 10 m resolution, Planet's PlanetScope at 4.8 m resolution, and Airbus SPOT at 1.5 m resolution. All images in South Africa were taken in February 2020, France in September 2019, and India in October 2020. As fields get smaller, higher resolution satellites are needed to delineate field boundaries.

The second major challenge is the lack of labeled data for training and validating models to delineate smallholder fields. As previously mentioned, ground- or aerial survey-based field boundaries are scarce in low-income regions. This leaves manual annotation of satellite imagery as the most viable source of boundaries [23–25,33]. Prior work in industrial agricultural systems used 480,000 labeled fields in South Africa [23] and 72,000 labeled fields in Australia [23] to train deep learning models at country-wide scales. Assuming the first challenge is addressed and one can access very high resolution imagery for generating labels, the small size of smallholder fields still poses a challenge: the smaller the average field size, the more fields there are to label in each satellite image. Since all deep-learning-based field delineation methods to-date require fully segmented boundary labels to supervise model training [21,23,28,31,32], creating labels in smallholder systems can become very costly. For example, suppose we want to generate segmentation labels for 1000 images. In a region where the average field size is 1 hectare, at PlanetScope resolution (4.8 m) a typical 256 × 256-pixel image would contain 150 fields. Labeling 1000 images would equal labeling 150,000 fields. The labor required to create fully-segmented labels that sample a representative fraction of a state, province, or country is therefore quite large, pointing to a need for more efficient ways of generating field boundary labels.

In this work, we explore how these two challenges can be overcome to accurately delineate field boundaries across India. The rest of this paper is organized as follows. In the

Dataset Section, we present the high-resolution satellite imagery and field boundary ground truth used to train and validate our deep learning models. We experiment with using both multi-temporal but lower-resolution PlanetScope imagery and very-high-resolution but single-date Airbus SPOT imagery. To overcome the lack of field boundary labels in India, we generate a new dataset of 10,000 manually-annotated field boundaries across the country. Unlike the fully-segmented labels usually used to train delineation models, the labels we created are "partial labels": only a fraction of the fields in each image are labeled, allowing fields from more locations across India to be sampled for the same labeling budget. In the Methods Section, we describe the neural network architecture, how to train on partial labels, and post-processing to obtain field instances. We test whether a model trained on field boundaries in France can reduce the number of labels required in India to achieve accurate delineation. While prior work has transferred field delineation models between industrial agricultural systems with success [23], this is the first time that such models have been transferred to smallholder agricultural systems.

The Results Section describes how our best model using Airbus SPOT imagery can delineate fields in India to operational accuracy, and how training on partial labels in France reduces the number of labels needed to perform well in India. In the Discussion, we synthesize the significance of these results: by efficiently collecting partial labels and combining high resolution satellite imagery with transfer learning, we enable smallholder fields to be delineated at large geographic scales. We also summarize an approach to delineating fields in regions with no existing field boundary datasets and highlight settings where field delineation remains challenging. Our final contribution is the release of our dataset of 10,000 fields in India, along with the weights of the field delineation models.

## 2. Datasets

### 2.1. Sampling Locations for Datasets

#### 2.1.1. India

To determine where in India to download satellite imagery and label fields, we used the Geo-Wiki dataset as a guide. Geo-Wiki contains a random sample of land surface locations globally that have been verified by crowdsourced volunteers to be cropland [37]. We manually inspected Airbus SPOT images centered at Geo-Wiki locations until we obtained a dataset of 2000 clear images for labeling. About a fifth of inspected images were removed due to the location not actually being cropland or the image being too low in contrast for fields to be delineated (Figure A1). While some parts of India saw a greater chance for images to be rejected due to low contrast (e.g., along the coast, in very wet areas), the dataset still spans the entire country.

To split the images into training, validation, and test sets, we divided India into a $20 \times 20$ grid of cells (Figure 2). In this paper, "validation" refers to the set on which hyper-parameters are tuned, while "test" refers to the set used to assess models' generalization performance. We assigned 64% of grid cells to the training set, 16% to the validation set, and 20% to the test set. All images that fell into a grid cell were assigned to the corresponding dataset split. Splitting images along grid cells minimizes the chance that images across folds contain the same or very similar fields, thereby preventing classification metrics from being inflated due to leakage.

#### 2.1.2. France

France is one of a handful of countries with publicly-available field boundary data. We use data in France to study the conditions needed for successful field delineation, and whether field delineation models trained in high-income regions can transfer to smallholder regions.

To construct a dataset of satellite imagery and field boundary labels in France, we first sampled 10,000 geographic coordinates at random from France's land surface. For each coordinate, we defined an image of $256 \times 256$-pixels at 4.77 m resolution (PlanetScope

imagery resolution) centered at the coordinate. The satellite image and field boundary ground truth were then obtained for each tile (see Sections 2.2.2 and 2.3.2).

For model development and evaluation, we split the sampled locations into training, validation, and test sets. As in India, France was discretized into a 20 × 20 grid of cells to minimize leakage among the splits. Each grid cell and all images that fell into it were placed in either the training, validation, or test set in a 64%-16%-20% split (Figure 2).

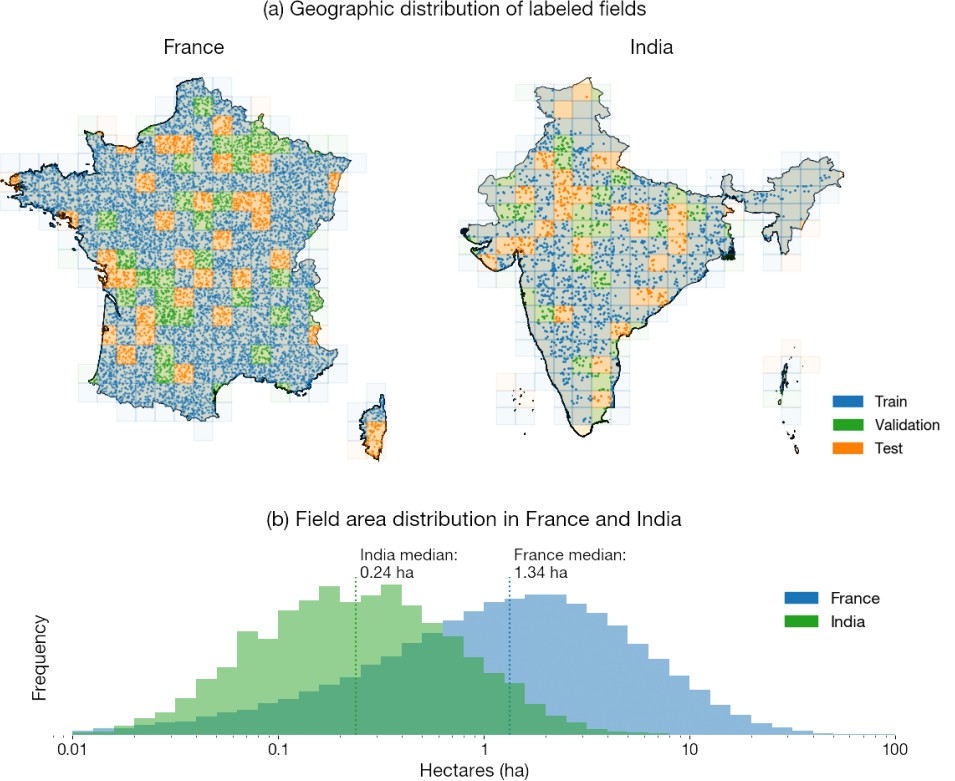

**Figure 2. Characteristics of France and India datasets.** (**a**) The locations of labeled fields are plotted as points on maps of France and India. We also visualize how images are split into training, validation, and test sets in geographic blocks. (**b**) Histograms of field areas on a log *x*-axis show that France fields are on average 5.6 times larger than India fields.

## 2.2. Satellite Imagery

### 2.2.1. Annual Airbus OneAtlas Basemap

The Airbus OneAtlas Basemap is a high-resolution map of Earth captured by the SPOT-6/7 satellites [35,36]. The imagery is 1.5m resolution and captures four bands: red, green, blue, and near-infrared. One basemap is created every year by stitching together hand-selected images with marginal cloud cover and seasonal consistency (meaning that adjacent imagery will be seasonally contiguous). The cloud cover target is less than 5% in regular areas and less than 25% in challenging areas; in practice, this means that most Airbus basemap images in India were taken during the dry season from October to March. We used the Descartes Labs platform to access Airbus imagery. Due to the Airbus terms of use barring raw bands of the imagery from being exported off the platform, we combined the 4 bands into 3 bands, creating a false color image that can be intuitively interpreted by human annotators. For more details, please see Appendix A.1.

Human annotators used Airbus SPOT imagery to label fields in India (see Section 2.3.1). We also used Airbus SPOT imagery as a model input for field delineation. We tried both Airbus SPOT imagery at native resolution (1.5 m) and down-sampled 2× and 3× (4.5 m) to compare against PlanetScope imagery. We downloaded Airbus SPOT images of 1500 × 1500 pixels at native resolution; at training time, a random 256 × 256 crop of

the image was taken, which allowed the partial labels to fall anywhere in the image and expanded the effective training set size (Figure A2).

### 2.2.2. Monthly PlanetScope Visual Basemaps

We also explored using monthly PlanetScope Visual Basemaps, which have a ground sampling distance of 4.77 m [38]. A distinct Visual Basemap is generated globally for each month using Planet's proprietary "best scene on top" algorithm, which selects the highest quality imagery from Planet's catalog over the course of the month based on cloud cover and image sharpness. A major advantage of Visual Basemaps is that they already chose the least cloudy image in an area, which is helpful in the subtropics where clouds are frequent. However, a downside is that, unlike individual PlanetScope images, Visual Basemaps only offer RGB bands. Since differences in vegetation often appear in the NIR range, we expect future work that uses NIR bands to perform even better than the results shown in this paper.

In France, we downloaded April, July, and October 2019 Visual Basemaps for the 10,000 locations. These months were chosen to span the growing season; April and October in particular mark the beginning and end of the growing season, when the contrast between adjacent crop fields is likely to be largest due to variation in sowing and harvest dates.

In India, we obtained Visual Basemaps for each location with labeled fields (see Section 2.3.1). We sampled $512 \times 512$-pixel PlanetScope tiles around each set of fields to achieve a larger effective dataset size; similar to for Airbus SPOT imagery, a random $256 \times 256$-pixel crop of the tile was taken during training (Figure A2). At each location, we downloaded Visual Basemaps for each month between August 2020 and July 2021. We later determined (Results Section 4.4) which months' imagery was best suited for field segmentation.

### 2.3. Field Boundary Labels

### 2.3.1. Creating Field Boundary Labels in India

Since no large-scale, geographically representative dataset of crop fields exists in India to our knowledge, we employed human workers to annotate fields across the country. Existing field boundary datasets are assembled either through field surveys [21], farmer submissions [22], or manual inspection of aerial or high-resolution satellite imagery [23]. Due to the expensive and time-consuming nature of field surveys, as well as a lack of digital infrastructure to query farmers about their field boundaries, we opted for the third strategy.

In order to label fields in satellite imagery, we need imagery of high enough resolution for field boundaries to be clear to human annotators. Since neither Landsat, Sentinel-2, nor PlanetScope imagery are high enough resolution (Figure 1), we chose to use Airbus OneAtlas Basemap imagery (1.5 m resolution).

At 2000 Geo-Wiki locations, we pulled Airbus SPOT images and asked human annotators to delineate the 5 fields that fall closest to the center of the image, which was marked with an asterisk. The number 5 was chosen to balance a wider geographic coverage (the fewer fields per image, the more images sampled, and the more of India's diverse geography is represented in the dataset) with the cost of labeling (the fewer fields per image, the more time annotators spend switching between images, and the higher cost per field). Our results showed that this was a good decision, as simulations in France yielded better models when models were trained on more images but fewer labels per image (Results Section 4.3). The final dataset consists of 10,000 fields in India, which were then split into training, validation, and test sets (Table 1).

For each sample in India, the label generated by human annotators was a binary raster at the same resolution as Airbus SPOT imagery (1.5 m). A pixel had a value of 1 if it was inside a field that the annotator decided to label, and 0 otherwise. We converted these labels into georeferenced polygons so that they can be paired with any remotely sensed data source. Note that, despite being vectorized, the field polygons should be considered drawn to 1.5 m accuracy.

**Table 1. Number of images and fields in each split of the France and India datasets.** Training, validation, and test sets were split in geographic blocks, with the split being around 64%-16%-20% train-val-test.

| Country | Number of Images | | | Number of Fields | | |
|---|---|---|---|---|---|---|
| | **Train** | **Val** | **Test** | **Train** | **Val** | **Test** |
| France | 6759 | 1546 | 1568 | 1,973,553 | 459,512 | 430,462 |
| India | 1281 | 300 | 399 | 6421 | 1500 | 1996 |

Because fields in an Airbus SPOT image can have ambiguous boundaries—due to low image contrast, small field size, or image blurriness—we asked the annotators to label only field boundaries that are clear in the image. This may result in bias in the India field boundary dataset toward omitting fields that are too small or low contrast for humans to see in Airbus SPOT imagery. The evaluation of field segmentation models in this study can therefore be understood as comparing how well the models perform relative to human interpretation of satellite imagery, rather than true fields.

### 2.3.2. Registre Parcellaire Graphique

Field boundary labels in France come from the Registre Parcellaire Graphique (RPG) [22]. The RPG is a georeferenced database of all agricultural fields in France that receive aid under the Common Agricultural Policy (CAP) of the European Union. An anonymized version of the dataset is released publicly each year, and we accessed this dataset at https://www.data.gouv.fr/ (accessed on 10 December 2020) [22]. The entire 2019 database contains 9.6 million plots, each drawn to centimeter resolution [22]. Although the RPG does not include farmland not receiving CAP aid, in reality 95+% of French agricultural land is recorded in the RPG. Our 10,000 images sampled across France contain over 2.7 million fields from the RPG; Table 1 describes how they are split into training, validation, and test sets.

### 2.3.3. Rasterizing Polygons to Create Labels

For each image, we rasterized the field polygons to create the following three labels to train our neural network (Figure A3). Using these three labels to supervise multi-task learning has previously been shown to outperform using only one label [39].

- The "extent label" describes whether each pixel in the image is inside a crop field. Pixels inside a crop field have value 1, while pixels outside have value 0.
- The "boundary label" describes whether each pixel is on the boundary of a field. Pixels on the boundary (two pixels thick) have value 1; other pixels have value 0.
- The "distance label" describes the distance of pixels inside fields to the nearest field boundary. Values are normalized by dividing each field's distances by the maximum distance within that field to the boundary. All values therefore fall between 0 and 1; pixels not inside fields take the value 0.

## 3. Methods

### 3.1. Neural Network Implementation

The first step of our field delineation pipeline is to use a deep neural network architecture to detect field edges. The architecture, named FracTAL-ResUNet, was first proposed in Diakogiannis et al. [40], where it outperformed other architectures for satellite image-based building change detection. It was subsequently used in Waldner et al. [31] to create production-grade field boundaries from Sentinel-2 imagery in Australia. We experimented with the FracTAL-ResUNet, ResUNet-a [39], and regular U-Net [41] architectures and found that the FracTAL-ResUNet had the best performance in France and India.

We briefly describe the FracTAL-ResUNet architecture and loss function here and refer the reader to Diakogiannis et al. [40] and Waldner et al. [31] for more details. A

FracTAL-ResUNet has three main features, reflected in its name: (1) a self-attention layer called a FracTAL unit that is inserted into standard residual blocks, (2) skip-connections that combine the inputs and outputs of residual blocks (similar to in the canonical ResNet), and (3) an encoder-decoder architecture (similar to a U-Net). We used a FracTAL-ResUNet model with a depth of 6 and 32 filters in the first layer, following the architecture in Waldner et al. [31].

The FracTAL-ResUNet is trained on a Tanimoto with complement loss, which was introduced as a new loss function for supervising segmentation in Diakogiannis et al. [39]. The authors found that training on the Tanimoto with complement loss outperformed training on the more commonly-used Dice loss. For ground truth labels **y**, model predictions **ŷ**, and a hyperparameter $d$, the Tanimoto similarity coefficient is defined as

$$\mathcal{FT}^d(\mathbf{y}, \hat{\mathbf{y}}) = \frac{1}{2}\Big(\mathcal{T}^d(\mathbf{y}, \hat{\mathbf{y}}) + \mathcal{T}^d(1 - \mathbf{y}, 1 - \hat{\mathbf{y}})\Big) \tag{1}$$

where

$$\mathcal{T}^d(\mathbf{y}, \hat{\mathbf{y}}) = \frac{\mathbf{y} \cdot \hat{\mathbf{y}}}{2^d(\mathbf{y}^2 + \hat{\mathbf{y}}^2) - (2^{d+1} - 1)\mathbf{y} \cdot \hat{\mathbf{y}}} \tag{2}$$

The Tanimoto similarity coefficient takes the value 1 when $\hat{\mathbf{y}} = \mathbf{y}$, i.e., the predictions are perfect. To maximize the similarity coefficient during training, we train to minimize the loss

$$\mathcal{L}^d(\mathbf{y}, \hat{\mathbf{y}}) = 1 - \mathcal{FT}^d(\mathbf{y}, \hat{\mathbf{y}}) \tag{3}$$

The Tanimoto loss was shown in previous work [39] to result in more accurate boundary predictions than cross entropy loss. The FracTAL-ResUNet is trained to output three predictions: a field extent prediction, a field boundary prediction, and a distance to boundary prediction (Figure A3). Training on all three labels results in a more accurate field delineation than training on only one label (e.g., field boundaries). The Tanimoto loss is computed for all three predictions and averaged over them.

We trained all models until convergence (usually at least 100 epochs) with a learning rate of 0.001. Depending on GPU memory constraints, the batch size for our experiments ranged from 4 to 8 (when experimental results are directly compared, batch size was constant across experiments). During training, data were augmented with the standard practice of random rotations and horizontal and vertical flips.

### 3.2. Training on Partial Labels

Usually, labels used to train field delineation models [23,31] are fully segmented, which means that every pixel in an image is annotated. Fully segmented labels are costly to create, so we instead generated partial labels, where only a fraction of the image is annotated. This type of training strategy—using imperfect or partial labels—falls under the machine learning sub-field of weakly supervised learning [42,43]. We hypothesized that it is better to label fewer fields in more images than more fields in fewer images, with the rationale that sampling fields from more locations in a region leads to better generalization. Below we describe how we train models on partial labels and the experiment we conducted to test this hypothesis.

Masking out unlabeled areas

Training a neural network on partial labels instead of fully segmented labels requires changes to the training procedure. Instead of computing the loss at every pixel of the image, the loss is only computed at labeled fields and their boundaries. Unlabeled pixels are masked out (Figure 3); the model's predictions are not evaluated there. To implement masking, we created a binary mask for each partial label, where 1s correspond to labeled pixels and 0s correspond to unlabeled pixels. Before computing the loss, we multiply the predictions by the mask to set every pixel that falls outside of the mask to 0.

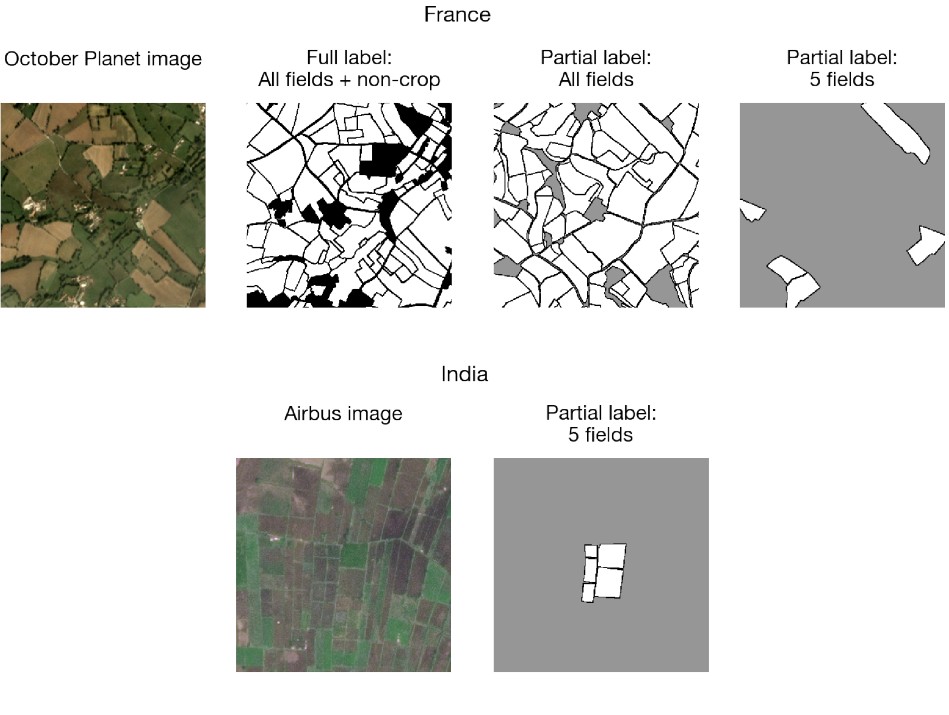

**Figure 3. Comparing full labels with partial labels.** Usually field delineation is trained on full labels, such as those available in France through the Registre Parcellaire Graphique. In this paper, we instead train on partial labels consisting of a labeled subset of fields and their boundaries. Unlabeled pixels are masked during training. Here we show examples of partial labels for France and India images, and denote masked pixels in gray.

Similarly, at evaluation time, model performance is evaluated on the labeled fields of the validation and test set images but not on unlabeled pixels. The performance at labeled fields should generalize to unlabeled fields as long as fields were labeled at random.

In France, we train one model on fully segmented labels and another on partially segmented labels that have all fields delineated but non-cropland areas masked. By doing this, we no longer task the model with classifying cropland versus non-cropland. Since prior work [33] has observed that errors in field delineation can arise from confusion between cropland and non-cropland, we hypothesize that decoupling these two tasks should increase accuracy on field delineation—especially when transferring a model across geographies.

In India, since annotators were asked to annotate 5 fields per image, the models are never trained on large swaths of non-crop labels, only on fields and the boundaries between fields. This means that, unlike models trained on fully segmented labels, models trained on these partial labels need a cropland map in post-processing to mask out non-crop areas. Given the availability and increasing accuracy of global cropland maps [44], this modular approach to field segmentation should be feasible.

Varying fields labeled per image

To optimize partial label collection, we simulated different annotation strategies using the fully-segmented France dataset. We assume that the total labeling budget allows for 10,000 fields to be labeled, but the number of fields per image and number of images labeled can vary. On one extreme, 125 training images were sampled across France and then 80 field labels were sampled per image. On the other extreme, 5000 training images were sampled and only 2 field labels were sampled per image. We also compared 200 images with 50 fields per image, 500 images with 20 fields per image, 1000 images with 10 fields per image, and 2000 images with 5 fields per image (Figure 3). All experiments were evaluated on the same validation and test set images, with partial labels of all fields in each

image (i.e., models were evaluated on all fields in each image and their boundaries, but not on non-crop areas).

### 3.3. Post-Processing Predictions to Obtain Field Instances

The deep learning model outputs field boundary detections but not separate crop fields. To obtain individual fields—also known as **instances**—we used a hierarchical watershed segmentation algorithm. Watershed segmentation is a region-based algorithm that operates on a grayscale image and treats it like a topographic map, with each pixel's brightness value representing its height. The algorithm separates objects within an image by seeding objects at local minima and then expanding them outward with increasing height (like water filling a drainage basin). Objects stop expanding when they reach the borders of other objects. Watershed segmentation has been shown to outperform other instance segmentation algorithms for field boundary delineation [23,45] and has been used in smallholder systems in prior work [21]. The specific implementation we used can be found in the Python package `higra` and is described in Najman et al. [46]. Both the field extent and field boundary predictions are required inputs to watershed segmentation. We tuned the algorithm's hyperparameters using validation set imagery.

We also point out that the FracTAL-ResUNet outputs predictions of size $256 \times 256$, but labels for Airbus SPOT imagery span windows larger than that size. To generate full predictions to be evaluated against the labels, we tile the Airbus SPOT imagery into overlapping $256 \times 256$ tiles, run each tile through the FracTAL-ResUNet, stitch the tile predictions into one image-level prediction, run watershed segmentation to obtain field instances, and then evaluate predicted instances against the ground truth labels.

### 3.4. Evaluation Metrics

We evaluate two outputs from the field delineation pipeline: (1) pixel-level predictions output by the FracTAL-ResUNet and (2) field instances obtained after post-processing.

Semantic segmentation metrics

Recall that the FracTAL-ResUNet outputs three predictions: field extent, field boundaries, and distance to field boundary. Each prediction is a raster of the same size and resolution as the satellite image input, with each pixel taking on values between 0 and 1. We evaluate only the extent prediction using overall accuracy, F1-score, and Matthews correlation coefficient. For partial labels, this involves evaluating the labeled fields with a pixel buffer around each field that serve as their boundaries.

The first metric is overall accuracy (OA), which is the most commonly used classification metric. We convert the model's field extent prediction to a binary prediction by setting values $\geq 0.5$ to 1 and values $< 0.5$ to 0. OA is then defined as

$$OA = \frac{TP + TN}{TP + FP + FN + TN} \tag{4}$$

where TP, TN, FP, and FN are the number of true positives, true negatives, false positives, and false negatives, respectively. When all predictions are perfect, OA $= 1$; when all predictions are incorrect, OA $= 0$.

The second metric is F1-score (F1), which is the harmonic mean of precision and recall. Mathematically, it is defined as

$$F1 = 2 \times \frac{\text{precision} \times \text{recall}}{\text{precision} + \text{recall}} = \frac{TP}{TP + \frac{1}{2}(FP + FN)} \tag{5}$$

The F1-score also ranges in value from 0 to 1, and requires both precision and recall to be high in order to be a high value.

The last and most discerning metric is the Matthews correlation coefficient (MCC). The MCC for a set of binary classification predictions is defined as

$$\text{MCC} = \frac{\text{TP} \times \text{TN} - \text{FP} \times \text{FN}}{\sqrt{(\text{TP} + \text{FP})(\text{TP} + \text{FN})(\text{TN} + \text{FP})(\text{TN} + \text{FN})}} \tag{6}$$

MCC measures the correlation between predictions and true labels, and is considered a more reliable metric than accuracy and F1 score [47]. When the classifier is perfect, the value of MCC is 1, indicating perfect correlation. When the classifier is always wrong, the value of MCC is $-1$, indicating complete negative correlation. MCC is only high if the prediction obtains good results in all four confusion matrix quadrants (TP, TN, FP, FN), and it does not produce misleading high values on very imbalanced datasets (unlike accuracy and F1-score). MCC was used in previous field segmentation studies to assess model performance [23,31], and we will use it as our primary metric for assessing field boundary detection.

When training models for many epochs, we keep the weights at the epoch with the highest validation set MCC. In our results, we report accuracy, F1-score, and MCC for each experiment on the test set, which is held out and never seen during model training or hyperparameter tuning.

Instance segmentation metrics

To evaluate the quality of predicted field instances after post-processing, we use the Intersection over Union (IoU) metric. IoU is a common metric for evaluating the accuracy of an object detector [48]. Given a set of ground truth pixels and a set of predicted pixels for an object, IoU is defined as

$$\text{IoU} = \frac{\text{Area of overlap}}{\text{Area of union}} \tag{7}$$

Perfect overlap yields an IoU of 1; no overlap, an IoU of 0. For each ground truth field, we compute its IoU with the predicted field that has the largest overlap. When evaluating over all fields in a dataset, we compute the mIoU and the fraction of fields with IoU greater than $k$%, which we denote as $\text{IoU}_k$. For example, $\text{IoU}_{50}$ is the fraction of fields where the overlap with a predicted field is at least half ($\geq 50$%).

*3.5. Field Delineation Experiments*

We conduct experiments varying input imagery and degree of transfer learning to optimize field delineation in India. The experiments are summarized in Figure 4 and described below.

PlanetScope vs. Airbus OneAtlas imagery

In France, we delineated fields using PlanetScope imagery since 4.8m is high enough resolution to clearly separate fields in France. In India, we conducted experiments with both PlanetScope and Airbus SPOT imagery as input, since Indian fields can be extremely small. Airbus SPOT imagery was also downsampled by up to a factor of 3 to approximate PlanetScope image resolution, in order to see whether differences between Airbus SPOT and PlanetScope performance stemmed from PlanetScope being lower resolution.

Combining multi-temporal imagery

Since we obtained PlanetScope mosaics for multiple months, we experimented with two different ways of combining multi-temporal inputs. In the first method, the model is fed multiple months of imagery separately. For example, in France the number of samples in the dataset became 30,000 and each label was replicated in the dataset 3 times (for April, July, and October imagery). We then followed the approach in Waldner and Diakogiannis [23] and averaged model predictions across the months to obtain a "consensus" prediction.

The second approach was to stack multiple months of imagery and feed them as a single input to the neural network. In France, the input became a 9-band image where the

first 3 bands were the April image, the second 3 bands were the July image, and the last 3 bands were the October image. We also experimented with shuffling the months in the stack at random for each sample. The rationale behind shuffling was that such a model would be more robust to India imagery looking different from France imagery, resulting in better performance when delineating fields in India.

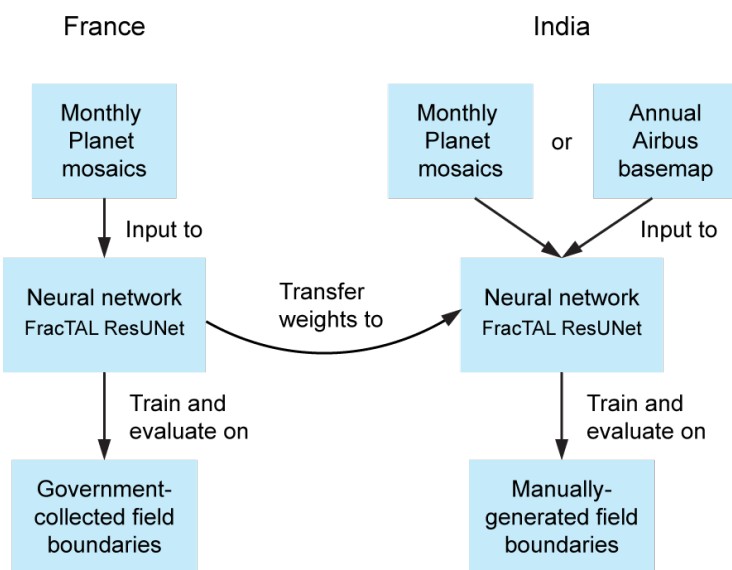

**Figure 4. Overview of field delineation experiments.** A model is first trained on imagery and field boundary labels in France before being transferred to delineate fields in India.

Downsampling France imagery

One potential challenge of transferring field delineation models across geographies is that fields vary in size and shape in different parts of the world. Fields in France are on average $5.6\times$ larger than fields in India (Figure 2).

To align France field sizes closer with India field sizes, we downsampled each PlanetScope image in France by $2\times$ and $3\times$, and trained a model on the union of these downsampled images. Note that a $2\times$ downsampling refers to downsampling $2\times$ along both the image height and width, which results in fields that are $4\times$ smaller in area. Downsampling causes France's field size distribution to overlap with India's field size distribution (Figure A4). We run experiments to see whether downsampling improves model transfer from France to India.

Training from scratch vs. transfer learning

Knowledge transfer from high-income regions with ample field labels could facilitate field delineation in smallholder regions. In this work, France serves as the source domain and India as the target domain. To test knowledge transfer, the following models were used to delineate fields in India:

1. A model trained on original-resolution PlanetScope imagery and fully-segmented field labels in France
2. A model trained on original-resolution PlanetScope imagery and partial labels in France
3. A model trained on 2x- and 3x-downsampled PlanetScope imagery and partial labels in France
4. A model pre-trained on PlanetScope imagery and partial labels in France, then fine-tuned on PlanetScope/Airbus SPOT imagery and partial labels in India
5. A model trained "from scratch"—i.e., without pre-training, starting from randomly initialized neural network weights—on PlanetScope/Airbus SPOT imagery and partial labels in India

The first three models simulate what happens when we have zero labels in India. Comparing model 1 to model 2 reveals any benefit of decoupling field delineation from cropland classification, and comparing model 2 to model 3 reveals any benefit of downsampling France imagery to match India field sizes. Comparing model 3 to model 4 shows the benefit of fine-tuning on India labels, while comparing model 4 to model 5 shows the benefit of pre-training on France labels.

We repeated the comparison of models 4 and 5 while varying the number of labels available in India for training/fine-tuning. Instead of using all 6400+ fields in the India training set, we restricted this number to 100, 200, 500, 1000, 2000, and 5000. For each training set size, a model was trained from scratch and compared to a model pre-trained in France and fine-tuned on the small training set.

## 4. Results

### 4.1. Field Statistics in India

We first describe the new field label dataset assembled as part of this study. The 10,000 fields in India ranged in size from 64 m$^2$ to 11 ha, with a median field size of 0.24 ha (Figure 2). Although we use very high resolution satellite imagery, the smallest fields are challenging to delineate even at Airbus SPOT resolution. Conversely, the largest fields can be delineated even with Sentinel-2 imagery. Most fields, however, fall in a range where they span at least tens to hundreds of pixels in PlanetScope and Airbus SPOT imagery, and it is therefore worth investigating whether they can be delineated successfully using these two types of satellite images. For instance, 90% of fields are larger than 0.057 ha, which corresponds to 25 PlanetScope pixels and 250 Airbus SPOT pixels. For comparison, field sizes in France according to RPG ranged from 3 m$^2$ to 2117 ha, with a median field size of 1.3 ha. France fields are therefore on average 5.6$\times$ larger than India fields. The smallest France fields are even smaller than the smallest India fields in our sample, while the biggest France fields are nearly 200$\times$ the size of the largest India fields.

Field sizes in India are not uniform across the country; fields are larger in northwestern and central India and smaller in eastern India (Figure A5). This is consistent with findings from the Geo-Wiki database [37], although our field sizes are more precise than the discretized bins in Geo-Wiki.

### 4.2. Interpreting Partial Label Results

Our results indicate that partial labels can successfully supervise field delineation (Tables 2–4). Before proceeding to interpret the results, we note that—due to class imbalance, the use of partial labels, and satellite images of different resolutions—the object-level metric IoU is a more straightforward indicator of performance than pixel-level metrics. In particular, because our labels consist of field interiors and field boundaries, there are many more "field interior" pixels than "not in field" pixels. Because of this class imbalance, accuracy and F1-score can appear to be high (values > 0.8) even when IoU is low (mIoU < 0.6). Meanwhile, MCC can appear low (MCC < 0.6) because of evaluation on partial labels even when IoU is high (mIoU > 0.8). Furthermore, we observed that PlanetScope-based predictions could have comparable pixel-level metrics to Airbus SPOT-based predictions, yet have lower IoU. An explanation of why this occurs can be found in Appendix A.2. The pixel-level results should therefore be interpreted in relative terms (when imagery and labels are the same in Table 3), while the object-level results an be interpreted in absolute terms (Table 4). Object-level results can also be compared to results in other papers.

### 4.3. Optimizing Partial Label Collection

For the same labeling budget, labeling fewer fields per image across more images improves model performance. In our France simulation, training on 125 images with 80 fields per image yields an MCC of 0.563, while training on 5000 images with 2 fields per image yields an MCC of 0.601 (Table 2). Between these two extremes, most of the advantage

of having more images is realized by 500 images (MCC = 0.596); adding more images beyond 500 only increases MCC slightly.

### 4.4. PlanetScope vs. Airbus OneAtlas imagery

In France, combining 3 months of PlanetScope imagery by stacking them into one 9-banded image performed better than taking the consensus of 3 separate predictions (MCC = 0.64 vs. 0.62, respectively). However, in India, the opposite was true; stacking resulted in slightly lower performance than taking the consensus (MCC = 0.48 vs. 0.52, respectively). Since our goal is to delineate fields in India, for the remainder of the results we combine PlanetScope imagery by taking the consensus prediction across months.

**Table 2. Partial label experiment in France.** We simulate a situation in France where we are constrained to only collecting 10,000 field labels to answer whether it is better to collect full labels for a few images or partial labels for many images. We vary the number of images from 125 to 5000 and the number of fields per image from 80 to 2, while keeping the total number of labeled fields constant at 10,000. The evaluation metric is the Matthews Correlation Coefficient (MCC).

| Number of Images | Number of Fields per Image | MCC |
|---|---|---|
| 125 | 80 | 0.563 |
| 200 | 50 | 0.585 |
| 500 | 20 | 0.596 |
| 1000 | 10 | 0.597 |
| 2000 | 5 | 0.601 |
| 5000 | 2 | 0.601 |

**Table 3. Pixel-level assessment of field delineation in France and India.** Table columns show results using a 3-month Planet image consensus in France and results using both the Planet consensus and Airbus imagery in India. Each row varies the amount of knowledge transfer from France and degree of additional training in India. The reported metrics are overall accuracy (OA), F1-score (F1) and Matthews correlation coefficient (MCC). The performance of the best model is highlighted in bold.

| Model | France | | | India | | | | | |
|---|---|---|---|---|---|---|---|---|---|
| | Planet Imagery Consensus (Apr, Jul, Oct) | | | Planet Imagery Consensus (Oct, Dec, Feb) | | | Airbus Imagery | | |
| | OA | F1 | MCC | OA | F1 | MCC | OA | F1 | MCC |
| Trained in France (full labels, native Planet resolution) | 0.89 | 0.88 | 0.78 | 0.74 | 0.83 | 0.32 | 0.84 | 0.91 | 0.17 |
| Trained in France (partial labels, native Planet resolution) | 0.91 | 0.95 | 0.50 | 0.79 | 0.87 | 0.35 | 0.87 | 0.92 | 0.38 |
| Trained in France (partial labels, downsampled Planet) | 0.89 | 0.93 | 0.62 | 0.76 | 0.84 | 0.39 | - | - | - |
| Pre-trained in France, fine-tuned in India (partial labels) | - | - | - | 0.82 | 0.89 | 0.52 | **0.90** | **0.95** | **0.51** |
| Trained from scratch in India (partial labels) | - | - | - | 0.81 | 0.88 | 0.48 | 0.91 | 0.95 | 0.50 |

**Table 4. Instance-level assessment of field delineation in France and India.** Table columns show results using a 3-month Planet image consensus in France and results using both the Planet consensus and Airbus imagery in India. Each row varies the amount of knowledge transfer from France and degree of additional training in India. The reported metrics are median Intersection over Union (IoU) and the fraction of fields with IoU over 50% ($\text{IoU}_{50}$). The performance of the best model is highlighted in bold.

| Model | France | | India | | | |
| --- | --- | --- | --- | --- | --- | --- |
| | Planet Imagery Consensus (Apr, Jul, Oct) | | Planet Imagery Consensus (Oct, Dec, Feb) | | Airbus Imagery | |
| | Median IoU | $\text{IoU}_{50}$ | Median IoU | $\text{IoU}_{50}$ | Median IoU | $\text{IoU}_{50}$ |
| Trained in France (full labels, native Planet resolution) | 0.67 | 0.63 | 0.32 | 0.24 | 0.30 | 0.18 |
| Trained in France (partial labels, native Planet resolution) | 0.70 | 0.68 | 0.37 | 0.33 | 0.74 | 0.69 |
| Trained in France (partial labels, downsampled Planet) | 0.65 | 0.62 | 0.32 | 0.35 | - | - |
| Pre-trained in France, fine-tuned in India (partial labels) | - | - | 0.52 | 0.52 | **0.85** | **0.89** |
| Trained from scratch in India (partial labels) | - | - | 0.50 | 0.50 | 0.85 | 0.90 |

When we trained on monthly PlanetScope images separately, the three months with the highest average MCC were October, December, and February. We therefore averaged these three months to obtain a PlanetScope consensus (Figure 5). We also tried a 6-month consensus (adding August, April, and June) as well as a 12-month consensus (August 2020 to July 2021). However, adding more months decreased the consensus performance. The best PlanetScope image-based result in India was a mIoU of 0.52 and an MCC of 0.52 (Tables 3 and 4). Inspection of instance predictions reveals that errors are more common as fields get smaller (Figures A6 and A7).

The results using Airbus SPOT outperformed results using PlanetScope; the best Airbus SPOT-based model in India achieved a mIoU of 0.85 and an MCC of 0.51. (Recall that, unlike IoU, the MCC results for Airbus SPOT imagery and PlanetScope imagery are not directly comparable.) Most of the superior performance using Airbus SPOT can be attributed to the higher spatial resolution of Airbus SPOT imagery (1.5 m) compared to PlanetScope imagery (4.8 m); when we downsampled Airbus SPOT imagery by 3× to 4.5 m resolution, the mIoU fell to 0.65 (Table A1). In addition, although PlanetScope imagery and 3× downsampled Airbus SPOT imagery are very similar in stated resolution, visual inspection suggests that downsampled Airbus SPOT imagery captures field boundaries more clearly (Figures 1 and 5).

For models trained on India data, Airbus SPOT predictions are more confident than PlanetScope predictions (i.e., there are more values closer to 0 and 1 rather than near 0.5; Figure 5); this could be explained by the PlanetScope prediction being a consensus of 3 images, which brings predictions closer to 0.5 when they disagree across months, as well as reflect the true uncertainty inherent to delineating small fields using PlanetScope imagery.

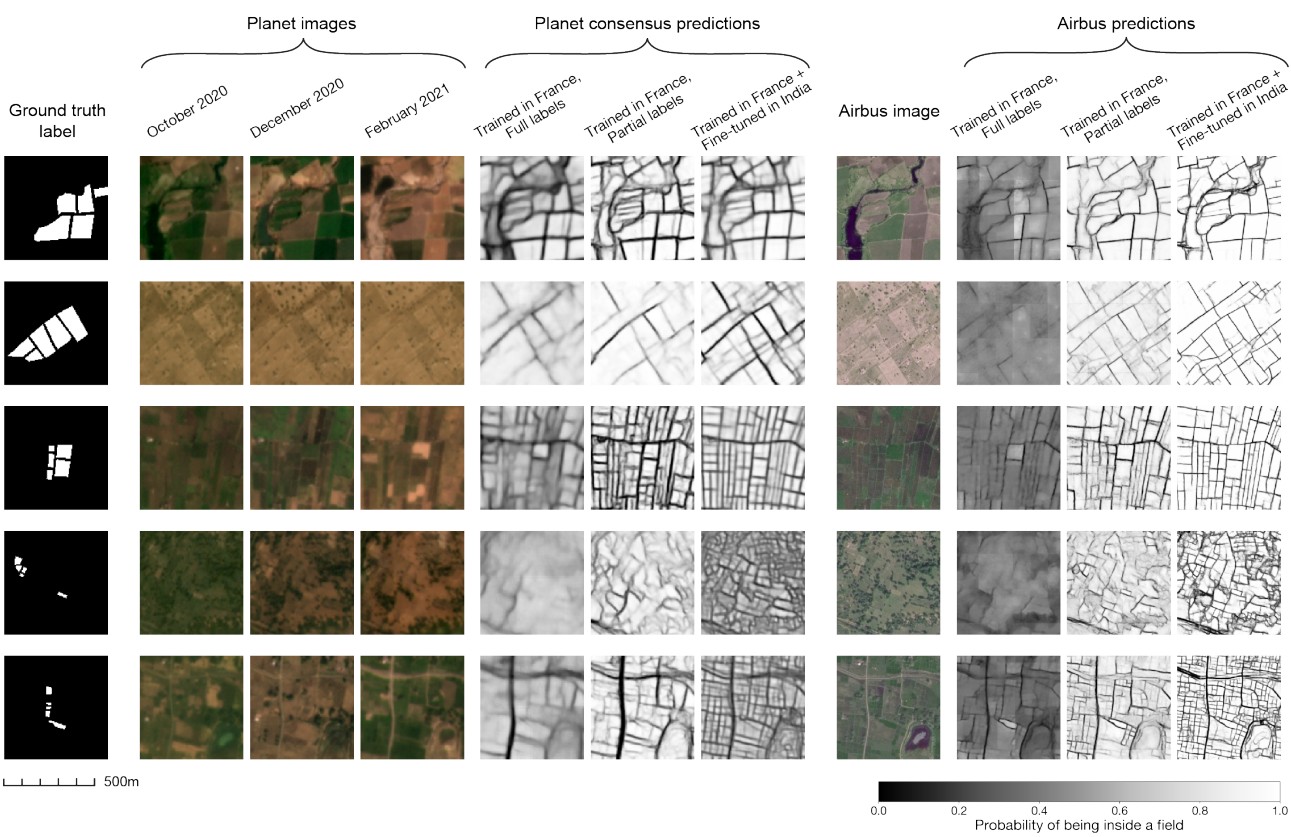

**Figure 5. Example model predictions in India.** We show the ground truth label, Planet Visual Basemap (PlanetScope) images for 3 months, Airbus OneAtlas Basemap (SPOT) images, and predictions using either Planet or Airbus images at five locations in India. Predictions are shown for three types of models: (1) trained only in France using full labels, (2) trained only in France using partial labels, and (3) pre-trained in France then fine-tuned in India using partial labels.

### 4.5. Transfer Learning from France to India

The rows of Tables 3 and 4 show the performance of models trained only in France, only in India, and first in France and then in India. Results are also shown for training on fully-segmented labels versus partially-segmented labels. Comparing these rows reveals the value of training on partial labels, the value of pre-training, and the value of collecting new ground truth labels for field delineation in India. Our findings are as follows.

#### 4.5.1. Training on Fully-Segmented France Labels Transfers Poorly to India

We trained our first model on fully-segmented field labels in France using PlanetScope imagery (Figure 3), following the common practice in existing literature [21,23,28,31,32]. This model achieves an overall accuracy of 0.89, MCC of 0.78, and mIoU of 0.67—performance that is higher than prior work in France [28] and similar to state-of-the-art work in South Africa [23] and Australia [31]. When we apply this model to India, however, the model fails to delineate fields: mIoU = 0.32 and MCC = 0.32 when using PlanetScope imagery and mIoU = 0.30 and MCC = 0.17 when using Airbus SPOT imagery. Upon examining the model's predictions in India, it appears that errors arise from not only failing to detect field boundaries but also mistaking cropland for non-cropland (Figure 5). The cropland versus non-cropland confusion is especially problematic when the model is applied to Airbus SPOT imagery; predictions on Airbus SPOT rarely detect field interiors with high confidence (and therefore appear dark in Figure 5).

#### 4.5.2. Changing to Partial France Labels Enables Transfer

By training on partial labels, we decouple the task of delineating fields from the task of differentiating cropland from non-cropland. Decoupling the two actually improves field

delineation in France slightly; mIoU increases from 0.67 when trained on full labels to 0.70 when trained on partial labels. (MCC becomes deflated by evaluation on partial labels; for an explanation, see Appendix A.2.)

Interestingly, the France model trained on partial labels delineates fields moderately well when directly applied to Airbus SPOT imagery in India without further fine-tuning (Table 4). The mIoU achieved by the model is 0.74, which is actually slightly higher than the model is performing within France (due in part, we hypothesize, to the prevalence of pasture and vineyard land cover types in France that are difficult to delineate). Inspection of model predictions shows that the model still struggles in regions in India where fields are very small or contrast between adjacent fields is low (Figure 5); in such settings it tends to miss field boundaries when they exist, although not as severely as when applied to PlanetScope imagery in India. Nevertheless, many fields in India can already be delineated by simply using a model trained on partial France labels.

When we apply the model trained on partial France labels to PlanetScope imagery in India, we find that performance is still low but higher than the model trained on full labels (mIoU = 0.37, MCC = 0.35). We then downsampled PlanetScope imagery in France to better match India field sizes, but found that model transfer does not improve (mIoU = 0.32, MCC = 0.39). Analysis of IoU curves shows that the model trained on downsampled imagery has more high-IoU predictions and more low-IoU predictions relative to the model trained on original resolution imagery, so that on average the two models perform similarly (Figure 6). Examination of model predictions reveals that, even when trained on downsampled PlanetScope imagery, the model from France fails to detect many lines that mark field boundaries, especially when fields are small or the region in the image is arid (Figure 5).

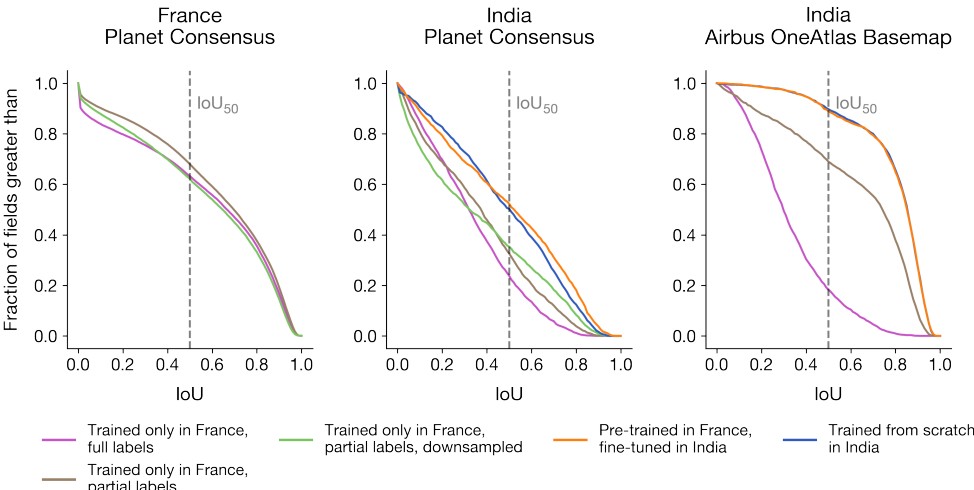

**Figure 6. IoU curves in France and India.** The *x*-axis shows the intersection-over-union (IoU) metric, and the *y*-axis shows the fraction of fields in the test set with an IoU above that threshold. Results are shown for five models: (1) trained only on France data (full labels), (2) trained only on France data (partial labels), (3) trained only on France data (partial labels, downsampled), (4) pre-trained on France data and fine-tuned in India, and (5) trained from scratch in India. Curves toward the upper right correspond to higher performance.

### 4.5.3. Training on Partial India Labels Improves Performance

Fine-tuning the models on India field labels improves performance substantially (second/third rows versus the fourth row of Tables 3 and 4). For the model using PlanetScope imagery, mIoU improves from 0.32 to 0.52 (MCC from 0.39 to 0.52), while for the model using Airbus SPOT imagery mIoU improves from 0.74 to 0.85 (MCC from 0.37 to 0.51).

Example predictions show that the improvement is especially dramatic in arid regions and for very small fields (Figure 5). A model trained in France will tend to transfer well to fields in India that look similar to fields in France, and transfer poorly to fields that look

different. Examples can be seen in the second, fourth, and fifth rows of Figure 5, where arid regions and small fields are poorly delineated by the model trained in France. Fine-tuning improves field delineation significantly on both PlanetScope and Airbus SPOT imagery by changing the model weights to recognize boundaries the way they appear in Indian agriculture. Concretely, many more boundaries are detected in the second, fourth, and fifth rows of Figure 5 after fine-tuning.

Next, we find that training on 6400 India labels from scratch yields similar results to transfer learning from France (fourth row versus fifth row of Tables 3 and 4). Training from scratch on PlanetScope imagery yielded a mIoU of 0.50 (MCC = 0.48), while training from scratch on Airbus SPOT imagery yielded a mIoU of 0.85 (MCC = 0.51). Analysis of IoU curves shows that the transfer learning and training from scratch curves are very similar to each other, especially in the case of Airbus SPOT imagery (Figure 6).

### 4.5.4. Pre-Training in France Improves Performance When India Datasets Are Small

At smaller training set sizes, however, the difference between transfer learning from France and training from scratch in India becomes substantial (Figure 7). With 5000 training labels, the difference between pre-training in France and training from scratch is small (mIoU of 0.492 vs. 0.486 for PlanetScope, 0.85 vs. 0.84 for Airbus SPOT). However, at 100 labels, this difference becomes large, at IoU = 0.44 vs. 0.29 for PlanetScope and 0.80 vs. 0.57 for Airbus SPOT. Indeed, the model pre-trained on France data and fine-tuned on 100 India labels performs better than the model trained on 500 India labels from scratch when using either PlanetScope or Airbus SPOT imagery. With pre-training in France, the decrease in performance as dataset size diminishes is quite gradual. A mere 100 fields in India is enough to raise mIoU of the Airbus SPOT model from 0.74 (directly applying a model trained in France) to 0.80 (fine-tuning on 100 fields in India). When datasets are small, pre-training in France reduces the number of India field labels needed to achieve a given performance level by between $5\times$ and $10\times$.

### 4.5.5. Most Errors Are Under-Segmentation Due to Low Image Contrast

To better understand the factors that affect field delineation, we visualize the geographic distribution of IoU and the relationship between IoU and field size for our best-performing model (Figure 8). We see that, while performance is high across India, IoUs are lower on average in east India than in west and central India. There is also a positive association between field size and IoU. When using Airbus SPOT imagery, it appears that it may be difficult to achieve IoUs higher than 0.8 when fields are smaller than 0.06 ha (equivalent to spanning fewer than 270 pixels in an image). Only 11% of fields in our sample in India are smaller than this threshold. When we conduct the same field size-IoU analysis for PlanetScope imagery, our best PlanetScope-based model has trouble achieving mIoU above 0.8 for field sizes smaller than 1 ha (equivalent to spanning fewer than 440 pixels in an image) (Figure A7). Unfortunately, 90% of fields in India are smaller than this.

When we visually inspect predicted field instances, we observe that high contrast between fields results in accurate field delineation (Figure 9). Errors, meanwhile, fall into two categories: under-segmentation (too few fields) or over-segmentation (too many fields). Most errors observed are under-segmentation that can be attributed to low contrast between fields (Figure 10); even though human labelers infer distinct fields from faint lines and nearby field boundaries, the model misses boundaries when they are too faint. Over-segmentation occurs when colors change significantly within one field, possibly due to variations in soil composition, vegetation health, and moisture levels.

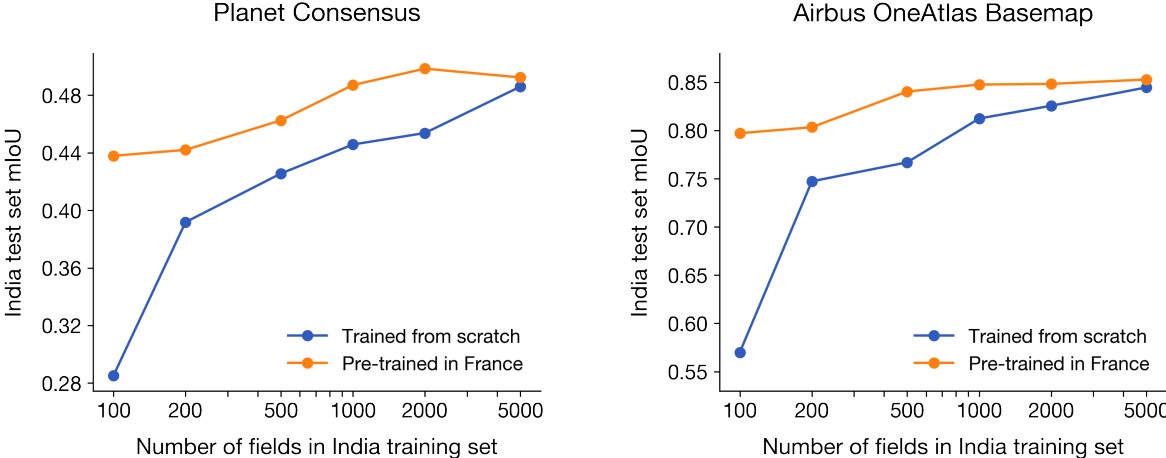

**Figure 7. The effect of pre-training a model in France.** We show the field delineation performance in India (measured by median intersection-over-union, mIoU) as we vary the number of labeled fields in the India training set for (left) a 3-month PlanetScope consensus and (right) Airbus SPOT imagery. As the number of training fields decreases, pre-training in France before fine-tuning on India fields confers more of a benefit over training a model from scratch on the India fields.

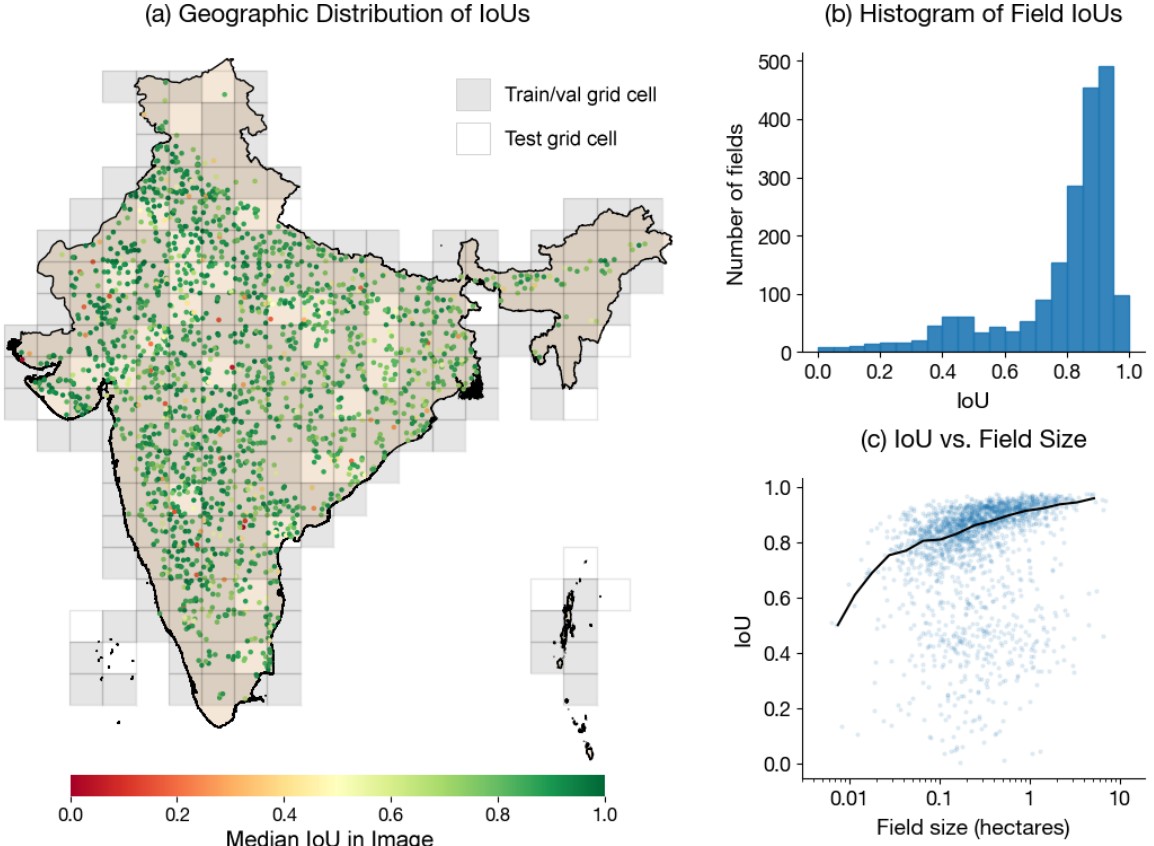

**Figure 8. Airbus SPOT model's prediction performance across India.** We visualize India-wide results for the model pre-trained in France and fine-tuned on Airbus imagery in India. (**a**) Mean IoU is plotted for each test set image over a map of India. Training and validation set grid cells are grayed out, while test set grid cells are not. (**b**) A histogram of field-level IoUs in the test set. (**c**) IoU and field size have a positive correlation. The black line shows the median IoU at a field size.

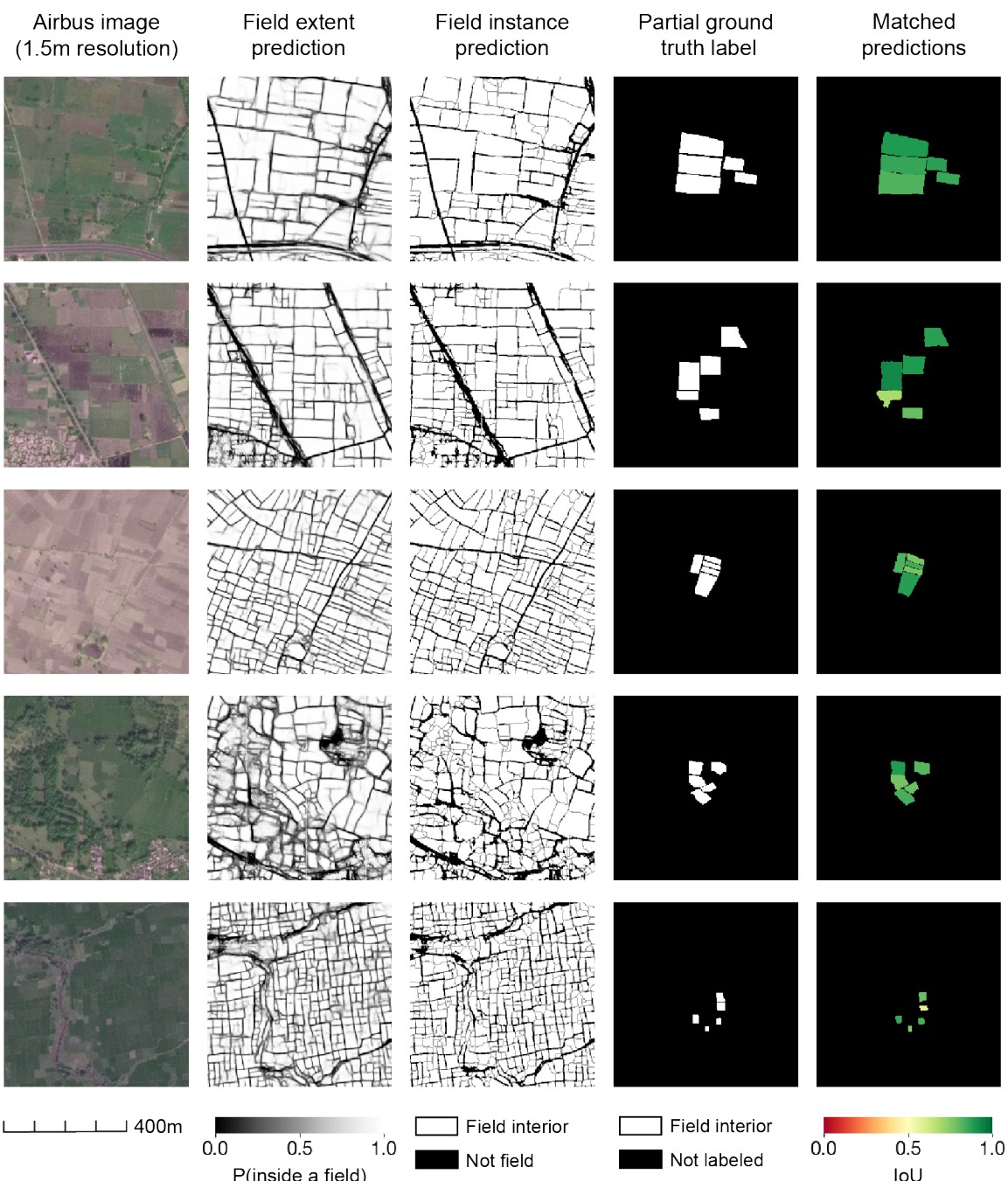

**Figure 9. Examples of using Airbus SPOT to delineate fields in India.** (**Left** to **right**) Airbus SPOT images in India and their corresponding field extent predictions, full field instance predictions, ground truth labels (5 fields each), and field instance predictions matched to the 5 labeled fields. For each matched field prediction, the color of the field corresponds to the IoU of the ground truth label with the predicted field.

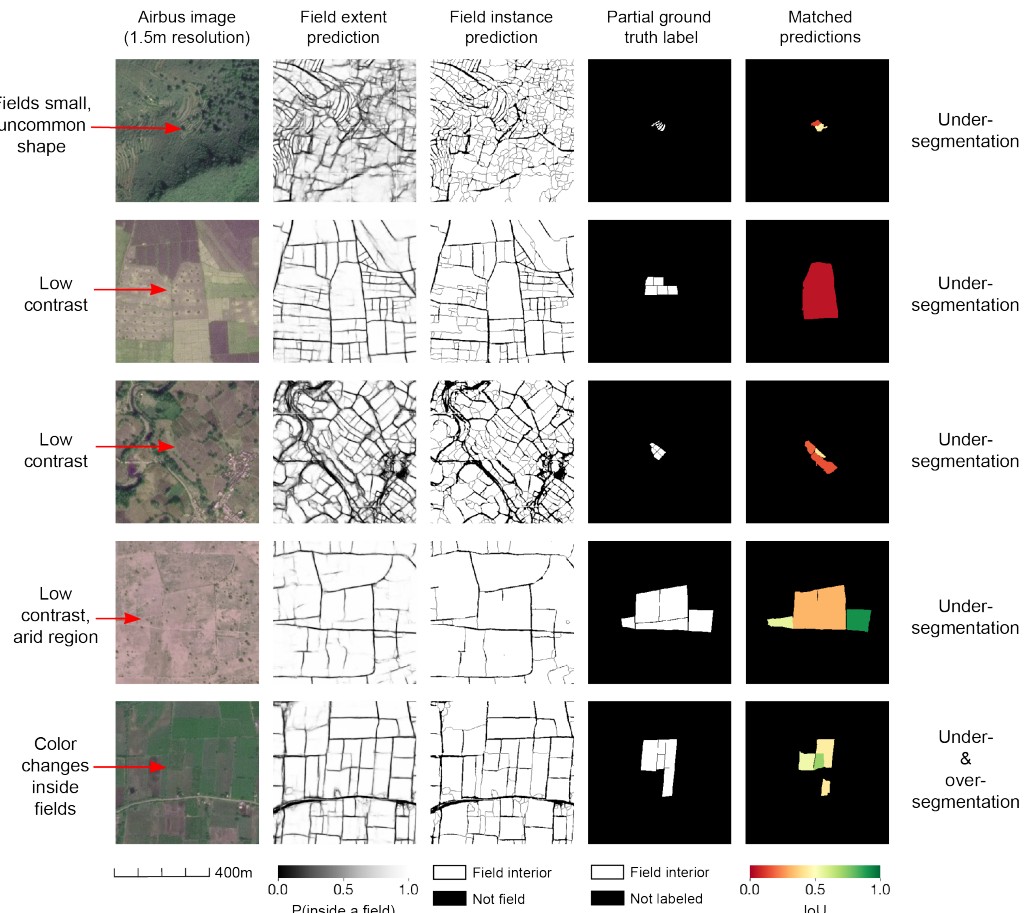

**Figure 10. Example images where the model performed poorly.** We show five examples that are among the 30 images with the lowest mIoU across fields. Most errors are due to under-segmentation attributable to low contrast between adjacent fields. Another source of under-segmentation appears to be uncommon field shapes or very small fields. Over-segmentation, which is less common, occurs when colors change significantly within one field.

## 5. Discussion

Few datasets of field boundaries exist in smallholder agricultural systems. In this study, we assembled a dataset of 10,000 fields in India and used high resolution satellite imagery, transfer learning, and weak supervision with partial labels to automatically delineate smallholder fields. Building upon prior work [23,31], our methods achieve high performance (best model mIoU = 0.85) through (1) access to very high resolution satellite imagery and (2) the use of partial field labels that relieve the burden of generating new labels, enable a large number of locations across a new country to be sampled, and facilitate model transfer from locations with existing labels.

In particular, training on partial labels decouples field delineation from cropland mapping, since non-crop areas are masked out when the model is trained on partial labels (Figure 3). Our experiments suggest that a model trained on fully-segmented labels in France can mistake cropland for non-cropland in India, causing field delineation to fail. Perhaps this is not surprising, since agriculture is quite different in France versus in India. Fields in France are on average 5.6× larger than fields in India, crop types and soils differ between the two countries, and France is dominated by a temperate oceanic climate while India spans tropical, temperate, and arid climates—these differences change the appearance of fields in satellite imagery and may reduce model performance when transferring a model trained in France to India. However, when trained instead on partial labels, we find that a model from France and applied as-is in India achieves a higher mIoU of 0.74. Furthermore, our simulation experiment in France shows that labeling a few fields in many images leads

to better performance than labeling many fields in few images. It is therefore not only possible but desirable to train field delineation models on partial labels, especially if the model will be transferred to other geographic regions. However, this method does assume that a separate model or product exists to classify cropland versus non-cropland. With the arrival and improvement of products like the Copernicus Global Land Cover Layers [44] and ESA WorldCover 10 m [49], this assumption is increasingly being met around the world. For field delineation using Airbus at 1.5m resolution, we expect higher resolution land cover maps like WorldCover to require less post-processing, but the best land cover map to use in a given region will depend on both cropland classification accuracy and resolution.

While a model trained in France can perform moderately well in India, fine-tuning with 6400 field labels in India improves mIoU to 0.85. This improvement is expected, since some fields in India lie outside of the France training set distribution and are therefore difficult for a model trained only in France to delineate well: for instance, northwest India is more arid than any region in France, eastern India is wetter than any region in France, and fields in India are smaller and have different shapes than fields in France. The same high mIoU is matched by a model trained on those 6400 labels from scratch. However, at smaller dataset sizes in India, we find that transfer learning yields better results than a model trained from scratch; pre-training helps to reduce the number of India labels needed by between 5× to 10×. With only 100 fields labeled in India, a model pre-trained in France and fine-tuned in India achieved an mIoU of 0.80. For reference, prior work creating an operational field boundary dataset in Australia by training on 70,000 labeled fields achieved an mIoU near 0.8 as well [31].

Our findings support the following approach for delineating fields in a target region with no existing field boundary dataset:

1. Pre-train a FracTAL-ResUNet neural network on source-region imagery and partial field boundaries.
2. Obtain remote sensing imagery of the appropriate resolution to resolve fields accurately in the target region.
3. Create partial labels for a representative sample of fields across the target region.
4. Fine-tune the neural network on a training set of labels in the target region.
5. Evaluate the neural network on a test set of labels in the target region. Repeat Steps 3 and 4 until model performance is satisfactory.

On a first pass, the user can skip Step 4 and generate just enough labels in Step 3 to evaluate a model trained only in the source region, since sometimes such a model may already yield good results. Should results not be satisfactory, one can iterate on Steps 3 and 4 until performance improves sufficiently.

The experiments with transfer learning suggest that Step 1 (pre-training on source-region data) will greatly reduce the number of target-region labels required from Step 3 to achieve the user's desired performance level. This implies that, for the same labeling budget, fields across a much larger geographic area can be delineated with transfer learning than without. The methods described in this paper can therefore greatly speed up the development of field boundary datasets at large scales. There is also a "network effect" for field boundaries—once a region similar to the target region has labels, the target region needs many fewer labels for successful field delineation. The 10,000 India fields labeled for this work can also be used for pre-training and can therefore facilitate field delineation in other smallholder regions. Once all agro-ecological zones and field shapes have sizable field boundary datasets, future field delineation should progress quickly due to transfer learning.

Some of the above steps could vary depending on characteristics of the target region. For example, pre-training could be done with France data, another source region's data, or with labels pooled from multiple regions. Greater similarity between the source region and the target region should yield a larger performance improvement from pre-training; therefore, the source region should be chosen to match the agro-ecological zone and field shapes of the target region as much as possible. The more different the colors, field shapes, and other attributes of a target region from the source region, the more labels we expect

will be needed in the target region to achieve operational field delineation performance. Furthermore, while PlanetScope imagery was in general too low resolution to delineate fields in India, we did find mIoUs above 0.8 for fields larger than 1 ha. Based on our results, we recommend using Airbus SPOT imagery for fields above 0.06 ha and PlanetScope imagery for fields above 1 ha. When combining multi-temporal PlanetScope imagery, we found that taking the consensus of separate predictions performed better than stacking multiple months into one image in India, but the opposite was true in France. One possible explanation is that stacking requires more model parameters to be learned, which is worse for a smaller dataset like that in India but better for a larger dataset like that in France. Our recommendation is therefore to use the consensus method when the number of field labels for training is small. For future work, using PlanetScope imagery containing the NIR band may lead to better performance than what we observed in this work, although our experiments downsampling Airbus SPOT imagery suggest that resolution is still the main limiting factor in smallholder systems. For fields that are even smaller than 0.06 ha, even higher resolution satellite imagery (e.g., 0.3m DigitalGlobe imagery) will be necessary.

While our method expands the geographies around the world where field boundaries can be created, a few barriers may still prevent successful field delineation in some regions or by some users. First, high-resolution satellite imagery may not be available due to either lack of user access or cloudiness in the region of interest. Clouds are especially likely to pose a challenge in the tropics. In such settings, aerial imagery could be an alternative source of inputs, since airplanes and drones fly beneath clouds and aerial imagery has been used to delineate fields in the past [24]. We leave an investigation of how well field delineation models can transfer between satellite and aerial imagery to future work.

Second, in some regions it may be difficult even for humans to annotate fields. We encountered examples of this in India due to low contrast between fields in imagery (Figure A1). Indeed, errors could exist in our India field labels if adjacent fields look similar at the time the Airbus SPOT image was taken. One solution to low contrast could be to reference multiple satellite or aerial images during labeling to increase the chance that fields will appear distinct from their neighbors in at least one image. Instead of only generating labels using Airbus SPOT imagery, future work could use a combination of Airbus SPOT, Maxar WorldView, PlanetScope, Planet SkySat, and other imagery. Satellite or aerial imagery-based annotations could also be replaced or augmented by field survey-based field boundaries. Each of these alternate approaches would increase labeling time and costs while increasing confidence in the labels. Although annual Airbus SPOT basemaps are available on the Descartes Labs platform, multiple Airbus images within one year remain costly to access. Likewise, aerial surveys and ground surveys are also very costly. Furthermore, we would expect a model using remote sensing imagery to perform less well in settings that require surveys to clarify field boundaries. Nevertheless, where fields can be distinguished in satellite imagery by humans, we expect the above pipeline to work well.

## 6. Conclusions

To date, crop field delineation in smallholder systems has been challenged by the low accessibility of high resolution satellite imagery and a lack of ground truth labels for model training and validation. This work combines high resolution PlanetScope and Airbus SPOT imagery, a dataset of 10,000 new field instances, state-of-the-art deep learning, weak supervision with partial labels, and transfer learning to automatically delineate crop fields across India.

First, human workers labeled 5 fields per image across 2000 images from throughout India. By training models on partial labels instead of fully-segmented labels, we reduced the labeling burden, sampled a broader diversity of landscapes across India, and decoupled field delineation from cropland mapping. Second, a neural network was pre-trained on partial field boundary data in France, where a large dataset of government-collected field boundaries is available. Lastly, the model was fine-tuned to predict partial field boundaries in India using either PlanetScope or Airbus SPOT imagery. Results show that accurate field

delineation can be achieved with Airbus SPOT imagery, with our best model obtaining an instance-level mIoU of 0.85. Interestingly, a model trained on only France fields still performs moderately well in India, with a mIoU of 0.74. Meanwhile, PlanetScope imagery appears only suitable for delineating fields larger than 1 ha. Further experiments showed that pre-training in France reduces by $5\times$ to $10\times$ the quantity of field labels needed in India to achieve a particular performance level.

Our method offers a scalable approach to delineating fields in regions lacking field boundary datasets. We release the dataset of 10,000 India field boundaries and trained model weights to the community with the goal of facilitating further method development and applications like crop type mapping, yield mapping, and digital agriculture in under-resourced regions of the world. The dataset and model weights can be found at https://doi.org/10.5281/zenodo.7315090 (accessed on 7 November 2022).

**Author Contributions:** Conceptualization, S.W. and D.B.L.; methodology, S.W. and F.W.; software, S.W. and F.W.; validation, S.W.; formal analysis, S.W.; investigation, S.W.; resources, S.W. and D.B.L.; data curation, S.W.; writing—original draft preparation, S.W.; writing—review and editing, F.W. and D.B.L.; visualization, S.W.; supervision, D.B.L.; funding acquisition, D.B.L. All authors have read and agreed to the published version of the manuscript.

**Funding:** This work was supported by the NASA Harvest Consortium (NASA Applied Sciences Grant No. 80NSSC17K0652, sub-award 54308-Z6059203 to DBL) and a Google Cloud Credit Award from Stanford's Institute for Human-Centered Artificial Intelligence. Work by SW was partially supported by the Ciriacy-Wantrup Postdoctoral Fellowship at the University of California, Berkeley.

**Data Availability Statement:** Data containing 10,000 crop field boundaries in India and weights of the best-performing model can be found at https://doi.org/10.5281/zenodo.7315090 (accessed on 7 November 2022).

**Acknowledgments:** We thank Descartes Labs for improving researchers' access to high resolution satellite imagery, and Rose Rustowicz in particular for helping us learn how to use the platform.

**Conflicts of Interest:** The authors declare no conflict of interest.

## Appendix A

### Appendix A.1. Details about Airbus Imagery

We access Airbus OneAtlas basemaps through the Descartes Labs platform because it is the most cost-effective way to obtain very high-resolution satellite imagery at country-scale. However, one limitation of this access is that the Airbus terms of use prevent any raw SPOT imagery from being moved off of the Descartes Labs platform. This means that either all data labeling and modeling must be developed on the Descartes Labs platform or we had to create a derivative of the Airbus SPOT imagery to work with it off-platform.

Since it was infeasible to perform all data labeling inside the Descartes platform, we opted for the latter solution. We transformed the 4-band Airbus SPOT imagery (RGB and NIR) into a 3-band derivative from which the original image values could not be inverted. In the new image, band 1 was the average of the red and green bands, band 2 was the average of the red and NIR bands, and band 3 was the average of the green and blue bands. The result was a false color image that still showed vegetation as green due to the strong NIR signal from vegetation. The appearance of the false color imagery was intuitive enough that human annotators had no trouble labeling crop fields.

Our creation of false color imagery means that the Airbus SPOT bands are not the same as the PlanetScope RGB bands. Transfer learning from France PlanetScope to India Airbus SPOT therefore has to overcome distribution shift in the input bands. The fact that transfer learning performed well indicates that the model learned features that are robust to this shift, although it is possible that transfer would have been even better without the shift.

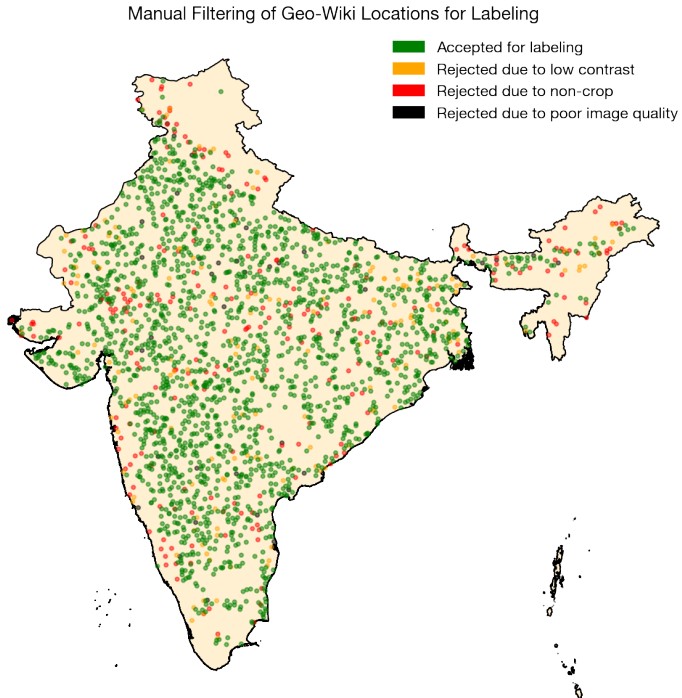

**Figure A1. Geographic distribution of locations accepted and rejected for field boundary labeling.**
The authors inspected 2446 Airbus SPOT images over Geo-Wiki locations and approved 2000 of
them for workers to annotate. The reasons for rejecting images were: image was too low contrast for
labeling, image contained no crop or very little crop area, and image was low in quality.

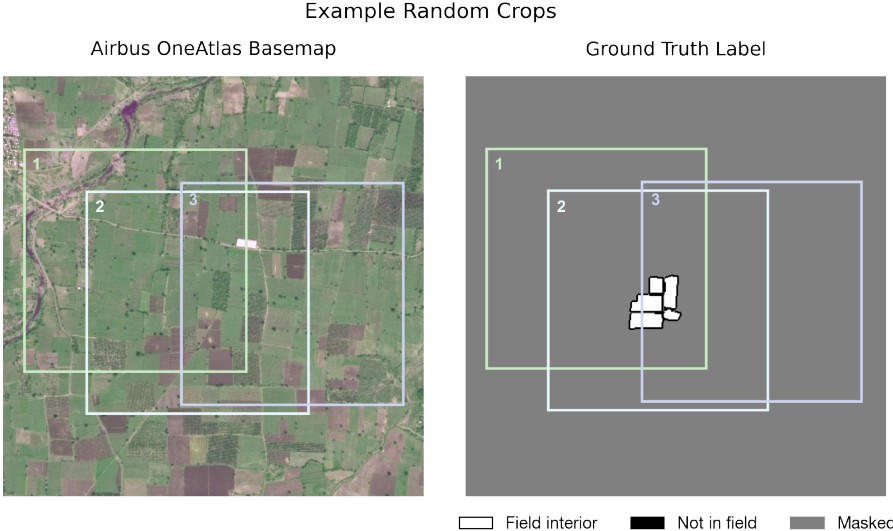

**Figure A2. Random image crops to increase effective dataset size.** We downloaded $512 \times 512$ pixel
images around the labeled fields in India; at training time, a random crop of the image was taken.
Compared to downloading $256 \times 256$ pixel images, this allows the model to see more diverse images,
expanding the effective dataset size and delaying overfitting.

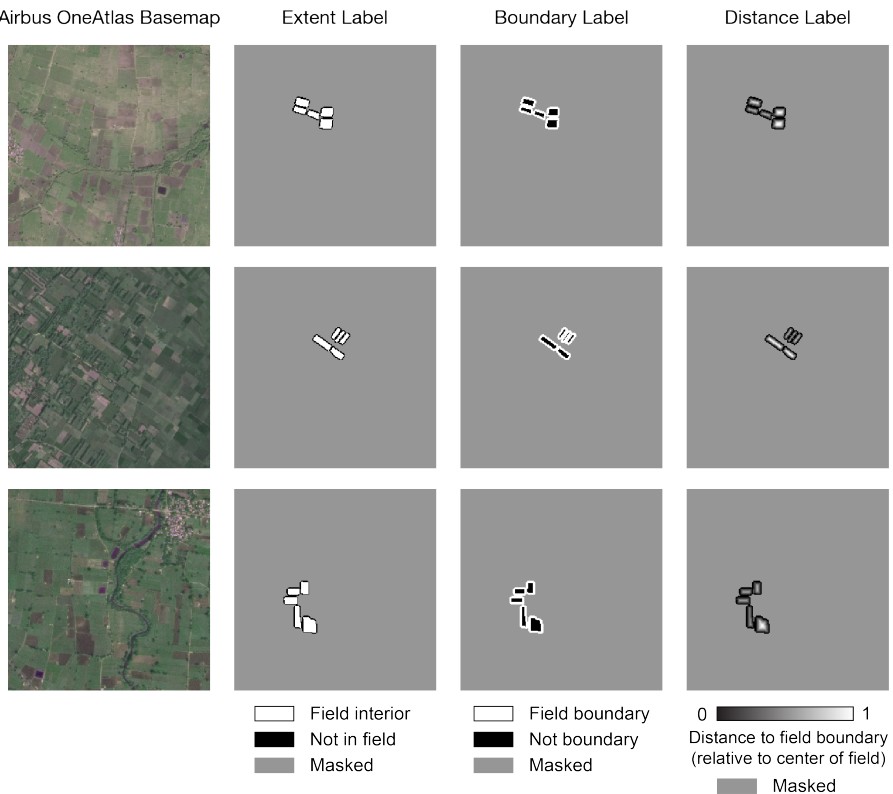

Airbus OneAtlas Basemap    Extent Label    Boundary Label    Distance Label

Field interior    Field boundary    0 ▬ 1
Not in field    Not boundary    Distance to field boundary (relative to center of field)
Masked    Masked    Masked

**Figure A3. Examples of labels used for multi-task learning.** Following the field delineation model design in Waldner et al. [31], we trained all neural networks in this paper to predict three tasks simultaneously: field extent, field boundary, and distance to field boundary. To train on partial labels, we mask out unlabeled pixels during training.

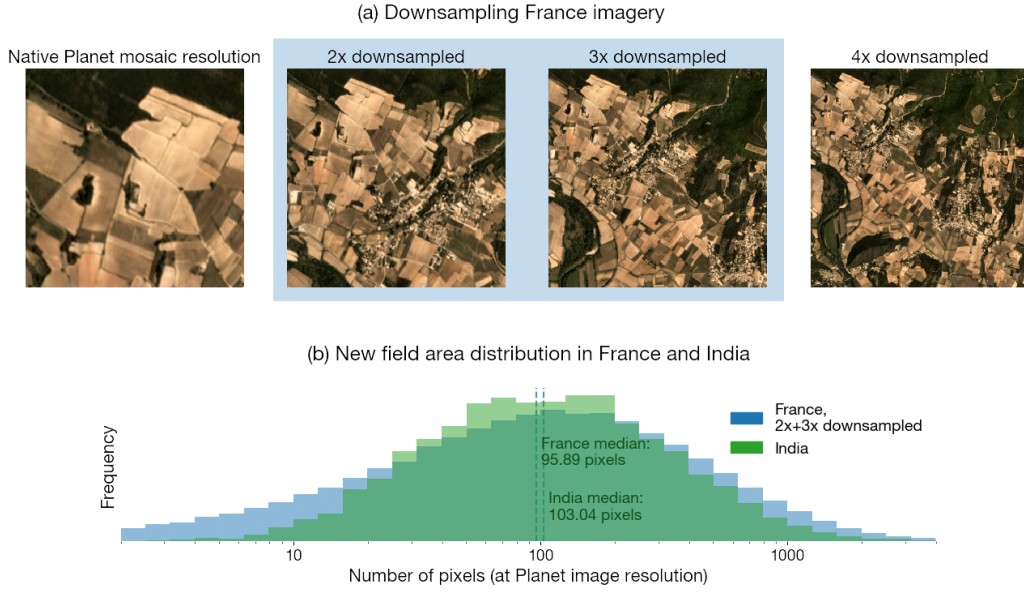

(a) Downsampling France imagery

Native Planet mosaic resolution    2x downsampled    3x downsampled    4x downsampled

(b) New field area distribution in France and India

France median: 95.89 pixels

India median: 103.04 pixels

France, 2x+3x downsampled

India

Number of pixels (at Planet image resolution)

**Figure A4. Downsampling France imagery to match Indian field sizes.** (**a**) We transformed the France dataset by downsampling PlanetScope imagery 2× and 3× and combining the two sets of images. (**b**) As a result, the transformed France field size distribution (as measured by PlanetScope pixels per field) matches the India distribution more closely than the original (Figure 2).

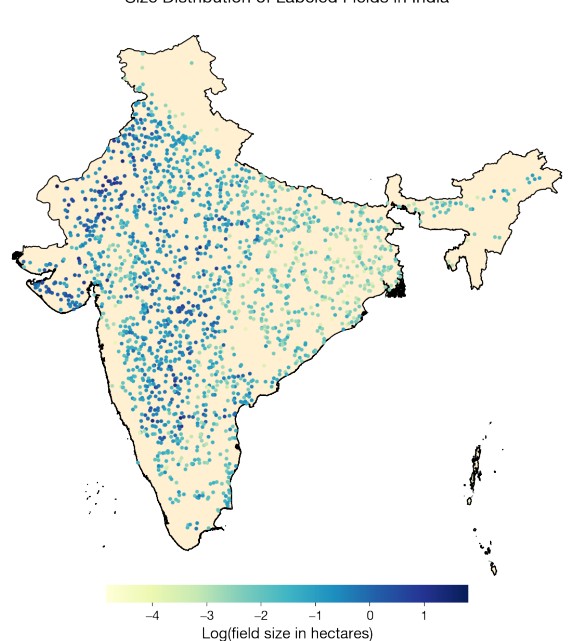

**Figure A5. India field size distribution from collected labels.** Each point represents a Geo-Wiki location where five field labels were collected. The average field size in each image is plotted on a log scale.

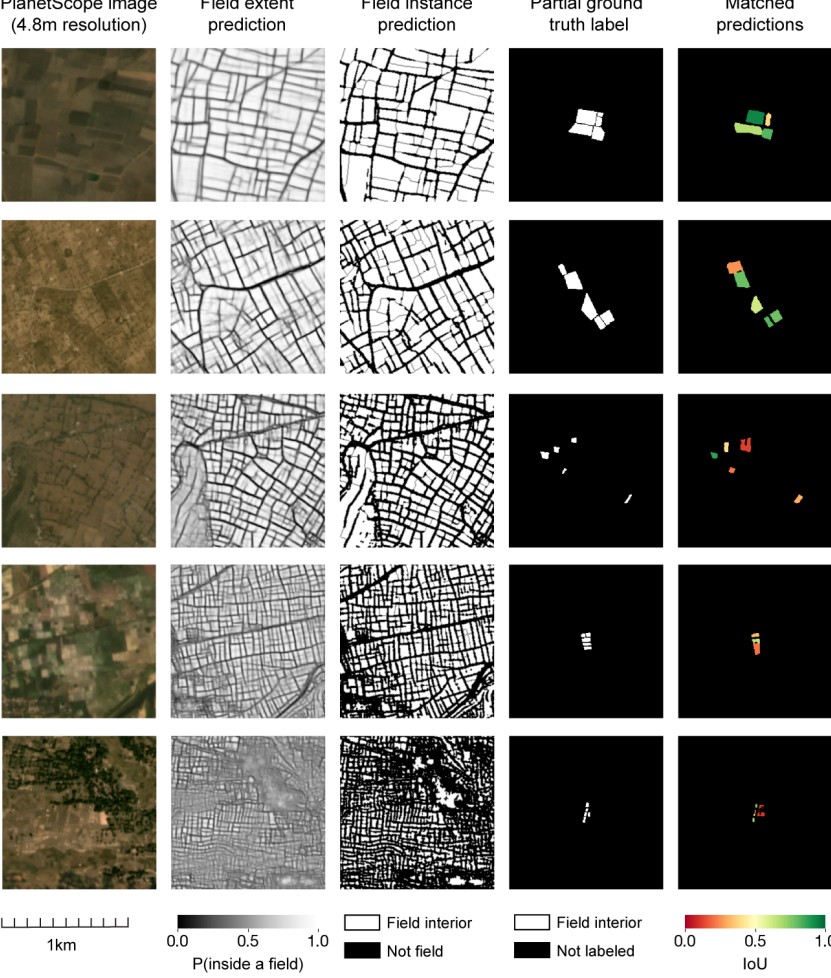

**Figure A6. Examples of using PlanetScope to delineate fields in India.** (**Left** to **right**) PlanetScope

imagery in India (October 2020) and their corresponding field extent predictions, full field instance predictions, ground truth labels (5 fields each), and field instance predictions matched to the 5 labeled fields. For each matched prediction, the color of the field corresponds to the IoU of the ground truth label with the predicted field. The lower IoU of PlanetScope predictions compared to Airbus SPOT predictions can be attributed largely to the lower resolution of PlanetScope, but may also in part be due to its lack of NIR band and errors in ground truth labels, which were generated using single-date Airbus SPOT imagery.

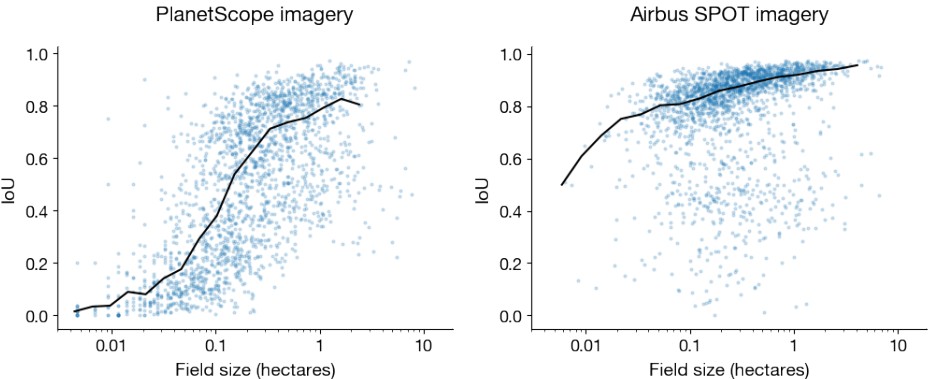

**Figure A7. IoU versus field size for fields delineated using PlanetScope and Airbus SPOT imagery.** Results are shown for the best model that employs each type of imagery. The black line shows the median IoU at a given field size.

*Appendix A.2. Interpreting Pixel-Level Metrics*

As mentioned in Section 4.2, pixel-level metrics can be difficult to interpret in field delineation when there is class imbalance, evaluation on partial labels, post-processing with the watershed segmentation algorithm, and comparison across satellite imagery of different resolutions. In particular,

1.  Accuracy and F1-score appear to be high (values > 0.8) because of class imbalance. Most pixels in our sampled images are inside a crop field rather than on the boundary, so just by predicting that all pixels are inside a field one can achieve a high accuracy or F1-score. In our results (Tables 3 and 4), an overall accuracy of 0.82 and F1-score of 0.89 correspond to a low IoU of 0.52 (Planet imagery in India). Since class imbalance is likely to exist in most field delineation problems globally, we caution against the use of accuracy and F1-score to interpret field delineation results.

2.  MCC is deflated when evaluated on partial labels instead of full labels; as a consequence, MCC values in this work appear lower than MCC values reported in prior work ([23,31]). Instance-level assessments, such as median IoU, do not suffer from this issue. For example, our FracTAL-ResUNet achieves an MCC of 0.73 when trained and evaluated on full field labels in France (Table 3), which is comparable to recent operationalized field delineation in Australia [31]. However, the same model trained and evaluated on partial field labels achieves an MCC of 0.50. In both cases, median IoU across fields is 0.70 (Table 4). The decrease in MCC is because full labels evaluate performance over non-crop area as well, and non-crop is easier to classify than field boundaries.

3.  Empirically, differences in accuracy, F1-score, and MCC reflect relative performance among experiments using the same imagery and labels. However, they do not capture the magnitude of differences in IoU when comparing across experiments using different imagery and labels. For example, in Table 3, the best Planet model achieves an accuracy of 0.82, F1-score of 0.89, and MCC of 0.52 in India. The best Airbus model achieves an accuracy of 0.90, F1-score of 0.95, and MCC of 0.51 in India. Comparing pixel-level metrics alone, the two models do not appear extremely different. However, the Planet model achieves an mIoU of 0.52 and the Airbus model achieves an mIoU of

0.85. The Airbus model is significantly better. To investigate why this might happen, we visualize images where MCC is higher on Planet imagery than Airbus imagery, but IoU is lower (Figure A8). One can see that MCC treats all pixels in an image equally, whereas watershed segmentation is highly affected by lines within a field. A thin line inside a field may only decrease pixel-level metrics slightly, but it can decrease IoU dramatically by breaking one field into two. Differences in label resolution may also play a role; Planet labels are lower in resolution than Airbus labels and our results suggest this may bias pixel-level metrics upward.

Since this work is concerned with delineating individual fields rather than separating crop vs. non-crop, we rely on IoU as the main indicator of performance. We report pixel-level metrics when comparing results for the same imagery and for continuity with existing literature.

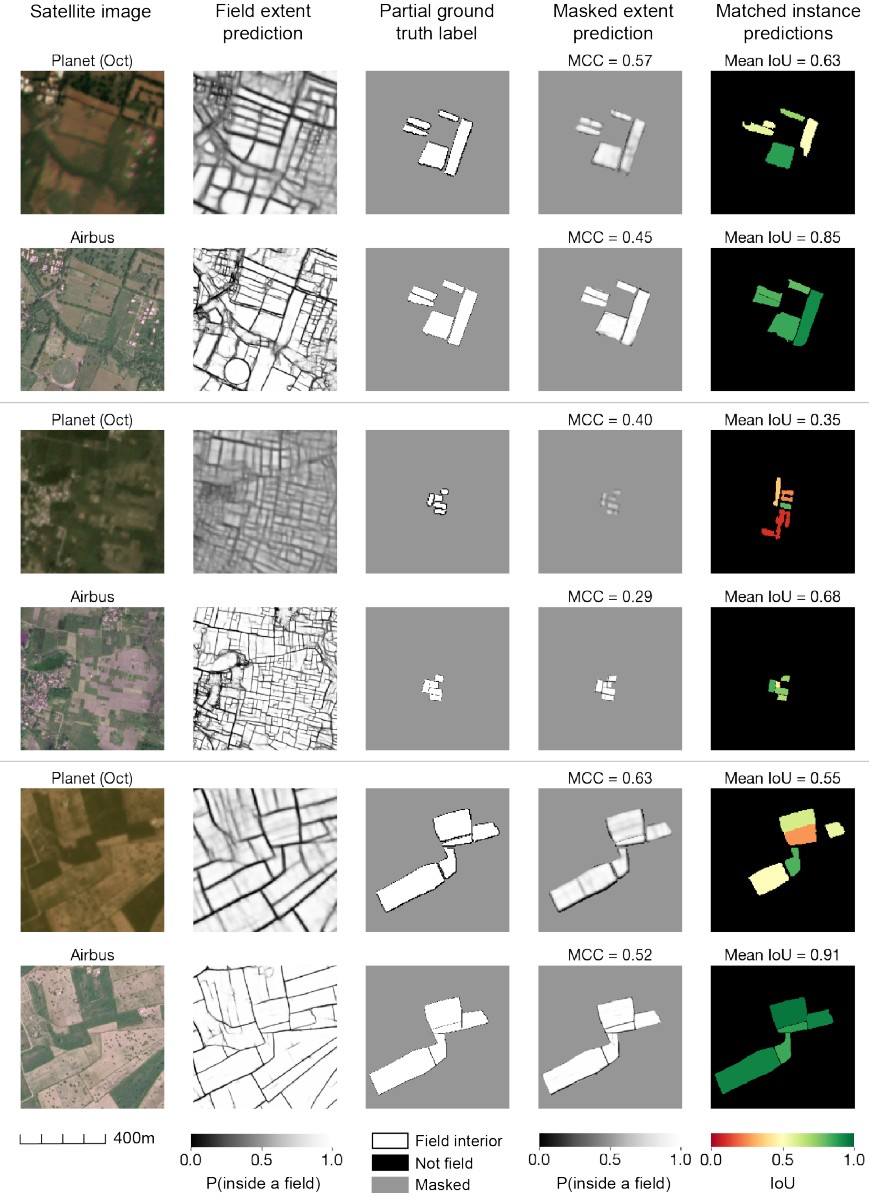

**Figure A8. Three example locations where Planet-based predictions have simultaneously higher MCC and lower IoU than Airbus SPOT-based predictions.** This illustrates that MCC is difficult to interpret across datasets with imagery of different resolutions and should be reserved for comparing experiments on the same dataset.

**Table A1. Downsampling Airbus imagery decreases field delineation performance in India.** Identical evaluation across the three rows allows us to isolate the effect of image resolution on field delineation performance. Performance declines moderately when Airbus imagery is downsampled by a factor of 2, and performance declines more steeply when Airbus imagery is downsampled by a factor of 3.

| Imagery | India | |
|---|---|---|
| | **mIoU** | **IoU$_{50}$** |
| Airbus full resolution (1.5 m) | 0.85 | 0.90 |
| Airbus 2× downsampled (3.0 m) | 0.80 | 0.88 |
| Airbus 3× downsampled (4.5 m) | 0.65 | 0.73 |

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
