# Peer review of "Unlocking Large-Scale Crop Field Delineation in Smallholder Farming Systems with Transfer Learning and Weak Supervision"

_remotesensing, doi:10.3390/rs14225738_

Round 1

Reviewer 1 Report

The manuscripts proposed a framework to delineate crop fields in smallholder farming systems with transfer learning and weak supervision. Transfer learning was undertaken by training a FracTAL ResUNet model using field boundaries collected in France and applying the model in predicting the field boundaries in India. Weak supervision was performed by training the model using partial labels within a satellite image instead of labeling all fields within the image. Five experiments were conducted to explore the benefit of (1) decoupling field delineation from cropland classification, (2) downsampling French imagery to match Indian field sizes, (3) fine-tuning on Indian labels, and (4) pre-training on French labels. Two types of satellite imagery were selected as base maps – Planet Scope Visual Basemaps (4.8m) and Airbus SPOT imagery (1.5m). Airbus imagery was downsampled to 3.0m and 4.5m to explore if the spatial resolution influenced the different results observed between Airbus and Planet. The performance of the models was evaluated at the pixel level by overall accuracy, F1-score, and Matthews correlation coefficient and at the object level by Intersection over Union. The best model achieved the Intersection over Union of 0.85 in India using 1.5m resolution Airbus imagery with pre-trained FracTAL ResUNet on France field boundaries and fine-tunes on India labels.

The manuscript was well-written and organized and was a pleasure to read. The experiments were clearly explained and properly designed and implemented. It is an interesting and useful publication that will benefit the readership. I only have some rather minor comments for consideration, which are listed below:

  • What does mIoU indicate in the abstract? Please clarify.

  • In some texts, median IoU was used, while in some texts, mIoU was used. Maybe use the same expression throughout the manuscripts.

  • Line 560. What does the ‘downsample Planet imagery’ mean? You mentioned two types of downsampling throughout the paper. One is the downsample of the Airbus to Planet resolution as in line #520. The other was the Planet image was downsampled by 2x and 3x to generate fields with smaller sizes. I believe it was the latter one. Please clarify to avoid confusion. 

  • Please clarify what MCC represents in Table 2. Similarly, please clarify what mIoU indicates in Figure 7. A table and figure should be standalone without referring to other text - like the other figures and tables in this manuscript.

  • Why was Mean IoU plotted in Figure 8(a), while median IoU was shown in Figure 8(c)? I suggest showing the Median IoU plot in Figure 8(a) to make the paper more consistent, considering median IoU was used in other parts of the manuscripts.
  • Why was Mean IoU used to process Figure 10 instead of median IoU? Please clarify—same suggestion as above.

  • In the introduction section, line #70 “Estes et al. [34] used unsupervised edge detection algorithms”, I believe they used Active Learning algorithm which is semi-supervised. I don’t think unsupervised edge detection is the correct way to describe their method. 

  • I think it would be more fitting to the flow of the paper if Figure 1 is moved after line #89.

  • Section 4.1 (or line #598-601) could be moved to the Dataset section 

  • In line #424, it is stated that the model can be applied on available cropland maps. Can it work on any cropland map? Or is there a range of spatial resolution where it can perform well? 

  • It would be interesting to understand how the model might accept: (1) satellite derived land cover products (with multiple classes such as cropland, forest, urban, water, etc.)? (2) unclassified satellite images? If so, how might that affect the performance? 

  • In line #605 of the Results section, there is a mention of class imbalance. What are the classes here? Cropland vs. Non-cropland? Labeled vs. Unlabeled? Weren’t the unlabeled pixels masked out before post processing? Can you elaborate more on what is the class  imbalance here?  

  • In the Discussion section, there is an explanation for how the method can be applied to regions around the world (with some exceptions). I’m curious to know how well this method will transfer to regions that have different shapes of crop fields. Would the high variance between field shapes limit the transferability of the model?

Author Response

Please see the attachment. Our responses are in blue.

Reviewer 2 Report

Dear Authors,

Thank you for your interesting work. My only remarks are related to:

the abstract that seems a bit long,

please place your figures in order closest to its first mention. I didn't find mentioning of fig 1 and 2, but firstly 3. Then 2 was mentioned in sec4. I think the ones you marked as A. should continue the normal numbering.

journal template,

Regards.

Author Response

We thank the reviewer for their time and consideration. Please see the attachment. Our responses are in blue.

Reviewer 3 Report

Dear authors and editor,

the manuscript “Unlocking large-scale crop field delineation in smallholder farming systems with transfer learning and weak supervision” proposes a new deep learning-based methodology for delineating field boundaries in the case of small fields. The authors aim to providing a methodology for delineating crop fields in regions of the world that lack field boundary datasets taking advantage of high-resolution Airbus SPOT satellite imagery. Moreover, the authors provided 10,000 new crop field boundaries for India. Although several methods and tools are already available for crop field delineation, the smallholders could benefit from the methodology presented here.

General comments:

The paper presents a huge labelling work that represents a step forward towards field boundary delineation across the world, but this does not mean the manuscript must be too lengthy. For example, in the Introduction there is a mixture of methodology and results that should be avoided.

The M&M are well described but they could be more synthetic (see specific comments)

The Results should be straightforward and easy to read. In this paper, there are too many speculations (that could be Discussion) and details that could be summarised in some tables.

I believe the topic here presented is worth to be published but I warmly suggest a reduction of the length of the paper eliminating redundant sections.

Lines 122-143: these are methods and don’t fit with the introduction

Lines 144-174: these are results and don’t fit with the introduction. You could also move them to the conclusions

Lines 180-184: this introduction could be avoided

Lines 337-344: this introduction could be avoided

Lines 613-617: these are not results, move to discussion

Lines 625-328: these are not results, move to discussion

Lines 634-639; these are not results, move to discussion

Lines 655-662: these are not results, move to discussion

Lines 685-697: these are not results, move to discussion

Lines 704-706: these are not results, move to discussion

Lines 74-744: these are not results, move to discussion

The table captions are usually reported before the tables

Author Response

(The authors gave the same response as above.)

Reviewer 4 Report

Crop field segmentation is one of the vital topics for remote sensing. The authors proposed a very interesting idea to utilize the available field annotations in France to smaller fields in India by applying the advantage of transfer learning. Using PlanetScope data to train a Spot 7-based model looks fine.  They applied FracTAL348 and watershed algorithms to solve their problem.  Algorithmic novelty is not very clear. However, I have some concerns, especially about the comparison.

You can find my comments below : 

About Compared Products:

The authors used the company names (Airbus and Planet) instead of the products (Spot7 and PlanetScope). I strongly recommend using product names during the comparison. Both companies have different products with higher/lower resolutions. Spot7 is a tasking-based satellite. Based on a given request, it focuses on a specific area, and as a result, it can acquire a high-resolution image. On the other hand, PlanetScope is designed to image the entire Earth's land surface every day. It can be a good idea to select similar product types during the comparison. (There are also some other tasking-based products which provide submeter resolution.)

About Selected Base maps:

According to Airbus ( https://storage.googleapis.com/p-oap-oaportal-assets/5ff87573e42c7f001b21ecce-OneAtlas_Basemap_Brochure.pdf ), OneAtlas Basemap provides only RGB images. However, the authors wrote that it was 4-band imagery. I recommend the authors double-check the used product name.

Similarly, Planet provides different base maps ( https://www.planet.com/products/basemap/ ). The authors used one of the products designed for human visual inspections (Visual Basemaps). As the authors mentioned visual maps include only RGB bands, it is a significant disadvantage for agricultural applications. I strongly recommend authors use Surface Reflectance Basemaps (or another analytics-ready product) during the training and testing of the models. Otherwise, the comparison will not be very fair.

About Annotations:

The authors tasked the annotators to label only field boundaries that are clear in the SPOT 6/7 image. As they mentioned, "This may result in bias in the India field boundary dataset toward omitting fields that are too small or low contrast for humans to see in Airbus imagery." In some cases, neighbor fields may have similar patterns(ex: during the dry season). So the annotators can draw the boundaries incorrectly. 

On the other hand, using monthly images, or just the correct month, may help us to detect the boundary of fields. Although the spatial resolution of the PlanetScope is lower, the temporal advantage may help a researcher to find these fields. However, the existing annotation/validation methodology will classify these fields as over-segmented. (I think Figure A.7 is an example case)

About Neural Network and  Transfer Learning:

Since the "Planet Visual Basemaps " and "One Atlas Basemap" have different characteristics, it requires some additional actions. However, it is unclear how the authors handled the normalization, the missing NIR band and other issues during the transfer learning. Can we get similar improvements if we use the same structure pretrained on a different dataset?

About Post Processing:

As I understand, the authors applied the watershed algorithm for both Planetscope and SPOT  7 images. Since the resolution of the Planetscope imagery is lower, the selected parameters can be critical. It would be great if the authors could show the instance predictions for the PlanetScope.

Author Response

(The authors gave the same response as above.)

Round 2

Reviewer 3 Report

Dear authors,

thank you for considering my comments. You did an effort to delete unnecessary sentences and be more concise. I think the topic may be very interesting and useful for the scientific community.

Reviewer 4 Report

Thank you for the updates.